cognition/psychology

abstract concepts, cooperative behaviour, joint action, kinematic indexes, social interaction

**Author for correspondence:**
Anna M. Borghi
e-mail: anna.borghi@uniroma1.it

†Equal contribution.

# Abstract concepts in interaction: the need of others when guessing abstract concepts smooths dyadic motor interactions

Chiara Fini[1,†], Vanessa Era[2,3,†], Federico Da Rold[4], Matteo Candidi[2,3] and Anna M. Borghi[1,4]

[1]Department of Dynamic and Clinical Psychology and Health Studies, and [2]SCNLab Department of Psychology, 'Sapienza' University of Rome, Italy
[3]IRCCS, Fondazione Santa Lucia, Rome, Italy
[4]Institute of Cognitive Sciences and Technologies, National Research Council (CNR), Rome, Italy

CF, 0000-0002-9939-4745; VE, 0000-0002-5288-1378

Abstract concepts (ACs, e.g. 'justice') are more complex compared with concrete concepts (CCs) (e.g. 'table'). Indeed, they do not possess a single object as a referent, they assemble quite heterogeneous members and they are more detached from exteroceptive and more grounded in interoceptive experience. Recent views have hypothesized that interpersonal communication is particularly crucial to acquire and use ACs. The current study investigates the reliance of ACs/CCs representation on interpersonal behaviour. We asked participants to perform a motor interaction task with two avatars who embodied two real confederates. Before and after the motor interaction task, the two confederates provided participants with hints in a concept guessing task associated with visual stimuli: one helped in guessing ACs and the other, CCs. A control study we performed both with the materials employed in the main experiment and with other materials, confirmed that associating verbal concepts with visual images was more difficult with ACs than with CCs. Consistently, the results of the main experiment showed that participants asked for more hints with ACs than CCs and were more synchronous when interacting with the avatar corresponding to the AC's confederate. The results highlight an important role of sociality in grounding ACs.

## 1. Introduction

Building abstract concepts (ACs), such as 'democracy' and 'justice', is a complex and sophisticated ability, and yet the use

of abstract words is ubiquitous [1]. Even if no concrete–abstract dichotomy exists, compared with concrete concepts (CCs), ACs have more heterogeneous members and no single object as a referent. Furthermore, they are more detached from exteroceptive and more grounded in interoceptive experience [2,3], and they are characterized by linguistic and contextual variability to a larger extent than CCs [4]. Finally, ACs are generally acquired at later developmental stages (age of acquisition) and more through linguistic explanations than through perception (modality of acquisition) [5–7] compared with CCs.

Recently, the issue of how ACs are acquired and represented has become increasingly debated (special topics: [8,9]). Until some years ago, the most influential theories on ACs learning/ representation were either embodied or distributional theories. According to embodied theories, both CCs and ACs were grounded in sensorimotor experience. Theories of distributed semantics, which intended meaning as given by associated words, ascribed a major relevance to language. Recently hybrid views, such as the multiple representation ones, emerged [10]. According to them, ACs would be grounded in sensorimotor systems (for recent evidence on the importance of visual and motor information, see [11]) like CCs, but they would activate to a larger extent linguistic, emotional and social experience. While there is plenty of evidence on the role played by both linguistic [12–15] and emotional experience [16] for ACs, social experience has not received the same attention.

Only very recently, some authors have started to investigate fine-grained differences among types of ACs without considering them as an indistinct whole [6,17–20]. Importantly, some recent studies have focused on the neural representation of abstract social concepts, comparing them with other concepts [21,22]. Within multiple representation theories on ACs, the words as social tools (WAT) proposal has put a special emphasis on sociality [23,24]. This emphasis is consistent with data showing that ACs, compared with CCs, evoke more introspective and social features [16,25,26].

According to WAT, sociality is crucial for acquisition, representation and use of ACs. The acquisition of ACs would involve linguistic and social experience more than the acquisition of CCs because the members of the latter are more heterogeneous and dissimilar. For example, it is more difficult to form the category of 'justice' than that of 'hammer' without others helping us in understanding its meaning (i.e. through social interactions and explanations).

Evidence has shown that ACs processing involves the activation of the mouth motor system [12,27,28]. The WAT theory proposes that such mouth activation might be due to different and possibly overlapping mechanisms. Here, we will focus on the 'social metacognition' mechanism [9,24]: during ACs processing, we would experience the metacognitive feeling that our knowledge is not adequate [29]. We would therefore prepare ourselves to ask for the help of authoritative others to complement it. A consequence of this would be that, if with ACs we need more help from others, we should be more cooperative when using them.

The present study aims at investigating the relationship between ACs and sociality. Specifically, it aims to test whether the social metacognition mechanism exists and influences the way we interact with our informants in a conceptual guessing task. In the first training phase, participants were shown images and had to guess the concept they referred to. They could ask for information from two confederates: one confederate gave them hints helpful to guess CCs, another to guess ACs. Participants were informed that these confederates could decide to be more/less helpful, giving them more/less useful hints. Then, participants performed a human–avatar motor interaction task, in which the avatar corresponded to the confederates' self-avatars (one associated with ACs and another to CCs, respectively). Finally, they underwent another guessing section with the two confederates.

Our idea is that being helped by another human to reach an intellectual objective might favour the subsequent attitude to establish a satisfactory sensorimotor interaction with him/her. Participants who have benefited from the other's help to guess an abstract meaning might develop the implicit willingness to create a successful relationship with him/her, knowing that the partner might help them in the following session.

We intend to verify three general hypotheses:

a) We expect participants to need more others' help in order to guess ACs than CCs.
b) We hypothesize that the attempt to gain the confederate's help leads to a more synchronous motor interaction with the ACs avatar. Previous studies have indeed shown that implicit positive attitude towards low-status individuals facilitates performance during interpersonal motor interactions [30].
c) We expect participants to show more automatic imitation of the ACs avatar's movements during the interpersonal interaction (i.e. visuo-motor interference effects). Indeed, it has been shown that the

automatic and unconscious imitation of other's movements creates a positive social relationship between interacting agents [31]. The human–avatar motor interaction task used in this study has been shown to elicit automatic imitation when participants perform complementary interactions both with a virtual and a human partner [32–35]. Using the same task, it has also been shown that automatic imitation is influenced by the social relationship between interacting people: participants imitate less their partner's movements when interacting with an out-group, if they have a negative implicit bias towards the out-group [35]. Here we hypothesize that, in the attempt to promote a positive social interaction, participants would imitate the other's action more. Establishing a positive relationship could be fruitful during the concept guessing task, in particular when participants will have to guess abstract concepts.

To sum up, we start from the assumption that, during a non-verbal social interaction, motion kinematics cues can indicate the implicit propensity to be better coordinated with the other. We thus expect participants to imitate more and be more synchronous with the movements of the AC avatar, in order to be helped in guessing the ACs. The goal to take advantage of the confederate's hints in guessing ACs should lead participants to be more coordinated and imitate more the corresponding avatar—in other words, to be more cooperative.

# 2. Method

All the hypotheses, experimental procedures and data analyses have been specified in a preregistration https://osf.io/4tbme. The analyses including the covariates and control experiments to assess the stimuli validity can be found in the electronic supplementary material. The paragraph named 'Stimuli validity check' in the electronic supplementary material contains analyses that were not preregistered and are therefore exploratory.

# 3. Participants

We tested 22 female participants and excluded one outlier (greater than 2.5 s.d. on the dependent variable grasping asynchrony (GAsynchr), see below), so that the final sample includes 21 participants (mean age 22.36, s.d. 2.46 years, mean education 14.45, s.d. 2.06 years). All participants had normal or corrected-to-normal vision. The study was in accordance with the Declaration of Helsinki and approved by the ethical committee of Sapienza University of Rome, Department of Dynamic and Clinical Psychology and Health Studies. Before starting the experiment, each participant was asked to sign the informed consent approved by the ethical committee of Sapienza University of Rome, Department of Dynamic and Clinical Psychology and Health Studies.

The choice of the sample size is given by reference to previous studies in the literature using the same task [35] and by a statistical estimation performed with the software More Power 6.0.4 [36]. More specifically, we have set as expected effects sizes the partial eta squared values obtained by [35] (0.4), where the same human–avatar motor interaction task was exploited to study the influence of the interactor's social identity on the ability to coordinate in the task, as in the present study. The output indicates that in a $2 \times 2 \times 2$ within factors design, a power of 0.95 and an eta squared of 0.4 [35], requires a sample size of 22 participants.

Participants were reimbursed 10 euros for their participation. They were recruited among university students by posting ads on social media, by asking one of their professors or by asking them directly to take part. Only right-handed female healthy participants were included, with normal or corrected-to-normal vision, who were Italian mother tongue. We tested only female participants in order to only include same gender pairs, as the confederates were two females. No other exclusion criteria were present.

# 4. Material

## 4.1. Concept guessing task

We selected a list of 40 words from the database of [5] (table 1). In order to prepare the material, we asked 10 university students who did not take part in the main experiment to produce six situations associated

**Table 1.** Dimensions of ACs and CCs selected words. Concrete and abstract concepts did not differ for familiarity, age of acquisition, word length dimensions.

| dimensions | abstract concepts | concrete concepts | t-test |
|---|---|---|---|
| familiarity | mean = 540.44 | mean = 502.92 | t(38) = 1.56, p = 0.12 |
| | s.d. = 59.7 | s.d. = 88.68 | |
| age of acquisition (AoA) | mean = 348.46 | mean = 326.61 | t(38) = 1.35, p = 0.18 |
| | s.d. = 48.75 | s.d. = 54.3 | |
| word length | mean = 7.7 | mean = 6.9 | t(38) = 1.18, p = 0.24 |
| | s.d. = 2.47 | s.d. = 1.71 | |
| abstractness | mean = 494.96 | mean = 154.02 | t(38) = 19.31, p < 0.001 |
| | s.d. = 67.8* | s.d. = 39.95 | |
| concreteness | mean = 262.03 | mean = 634.98 | t(38) = −23.98, p < 0.001 |
| | s.d. = 42.93* | s.d. = 54.69 | |
| modality of acquisition (MoA) | mean = 498.63 | mean = 300.65 | t(38) = 6.69, p < 0.001 |
| | s.d. = 82.04* | s.d. = 103.77 | |
| imageability | mean = 305.62 | mean = 634.98 | t(38) = −15.4, p < 0.001 |
| | s.d. = 76.8* | s.d. = 54.66 | |
| contextual availability | mean = 434.61 | mean = 593.64 | t(38) = −7.72, p < 0.001 |
| | s.d. = 68.88* | s.d. = 61.21 | |

with each word. We then found a corresponding picture for the situation that participants produced more frequently for each word.

Finally, we performed a pilot study asking 15 participants to indicate through a visual analogue 0–100 scale (VAS) (0 corresponded to not at all, 100 to extremely) to what extent they feel they would need another person's help in order to identify CCs and ACs associated with the selected pictures. Results showed that participants were keener (t(15) = −3.67, p < 0.001) to rely on others when they were required to identify ACs (mean = 83, s.d. = 15.33) than CCs (mean = 54.06, s.d. = 26.38).

## 4.2. Avatar of the joint-grasping task

The avatar of the joint-grasping task was the same used in previous studies [33–35]. It was created in Maya 2011 (Autodesk, Inc.) thanks to a customized Python script (Prof. Orvalho V., Instituto de Telecomunicacoes, Porto University), and the virtual scenario was realized in 3DS Max 2011 (Autodesk, Inc.). The avatar moved following the kinematics of a real actor's arm (SMART-D motion capture system (Bioengineering Technology & Systems (B | T | S))) that was recorded while performing 10 reach-to-grasp movements with the dominant right hand, five toward the upper part of the bottle (precision grip) and five toward the lower part (power grip). These 10 different movements (five precision and five power grips) were included in the experimental task as experimental stimuli. SMART-D motion capture system allowed to record the three-dimensional position of 19 passive reflecting markers, positioned on the right hand, forearm shoulder and chest (see [37], for further details). After recording the 10 different movements, SMART-D modules allowed to reconstruct and label the markers and to interpolate short missing parts of the trajectories. The final processed human-kinematics were realized using the commercial software MotionBuilder 2015 and 3DS Max 2015 (Autodesk Inc.) and implemented in a high-polygons three-dimensional model of a Caucasian male upper body. The avatar was displayed without its head, to avoid facial expressions having any unwanted influence. The duration of each clip (approx. 2000 ms) was the same for up (precision grip) and down movements (power grip). Each stimulus started with a still avatar, with its hands resting on the table. The avatar started its movement at a variable amount of time after the auditory go signal (i.e. between 200 and 500 ms). The moment in which it touched the bottle was computed by

(a)

(b)

(i)

(ii)

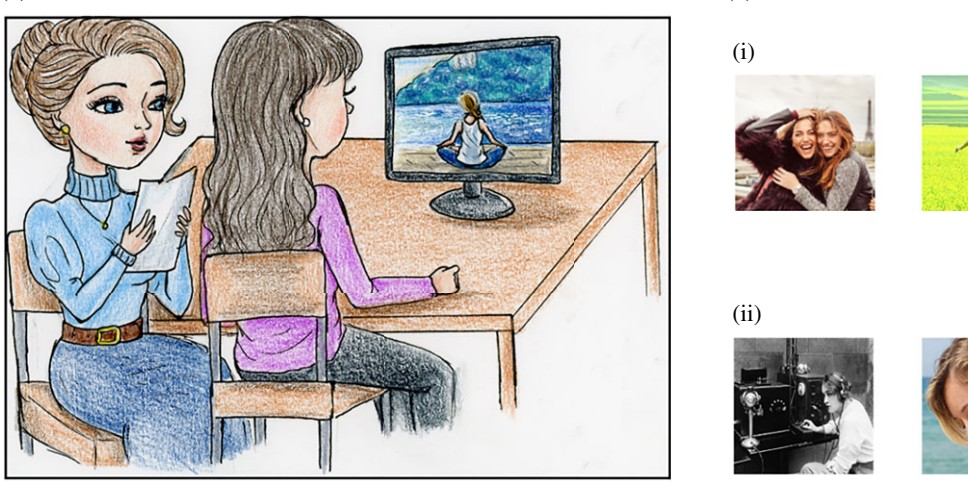

**Figure 1.** Illustration of the concept guessing task. (*a*) Participants were required to guess which concept the image refers. When they were not able to infer the concept immediately (i.e. after 60 s), they were allowed asking the confederates for suggestions. More specifically, participants would see an image, try to guess the associated concept; if not successful or if after 60 s they did not indicate any concept, they would receive a suggestion. Then they had to try to guess again and so on. In the drawing, the participant is trying to guess the AC 'peace'. (*b*) (i) Examples of abstract (friendship and freedom) and of (ii) concrete (telegraph and seashell) concepts.

having a photodiode on the screen displaying the videos that detected the appearance of a black dot positioned on the frame where the avatar touches the bottle [33–35,38].

# 5. Experimental tasks

## 5.1. Concept guessing task

Participants were presented with pictures referring to situations linked to CCs and ACs (e.g. 'muscle': 'a bodybuilder during the training'; 'freedom': 'to run on the grass', figure 1 for pictures examples) and were required to guess to which concept the image refers. When they were not able to infer the concept immediately (after 60 s), they were allowed asking the confederates for suggestions. The confederates could provide six suggestions taken from a list, corresponding to the most common situations associated with each concept indicated by the group of 10 participants mentioned before. Even if the participant guessed the concept without asking for all the six suggestions, the confederate would anyway read all the six suggestions, aiming to control for the amount of social interaction to which each participant was exposed. More specifically, participants saw an image and tried to guess the associated concept. If they were not successful or if after 60 s, they did not indicate any concept, they received a suggestion; then they had to try to guess again and so on. If, after receiving all the six suggestions, participants were not able to correctly guess the concept, the experimenter told them the correct concept. Only one confederate, the one in charge of delivering hints about ACs or CCs was physically present in the room during the guessing tasks; the other one was waiting her turn to take part in the following experimental session.

Importantly, participants were told that the confederate might decide whether to be very collaborative (providing more helpful hints first) or not very collaborative (providing less helpful hints first). We reasoned that such a remark would motivate participants to establish a fruitful relationship with the confederate, aiming to obtain from her more helpful hints.

The confederates associated with each set of concrete/abstract pictures were counterbalanced across participants.

## 5.2. Human–avatar motor interaction task

We exploited an ecological and well-controlled human–avatar interactive task; i.e. the 'joint-grasping task' [33–35,38], that represents the ideal paradigm to study an online dyadic interaction with a

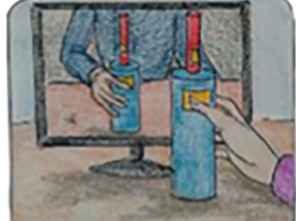

imitative trial

'grasp the bottle shape object
as synchronously as possible
with your partner'

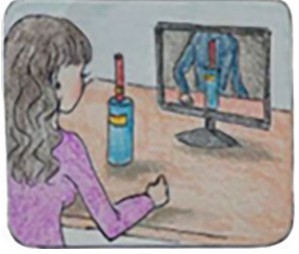
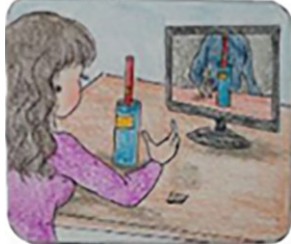

complementary trial

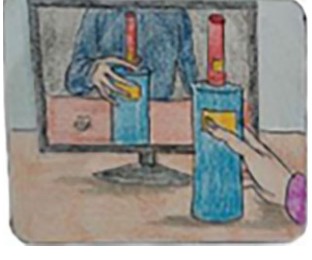

**Figure 2.** Illustration of the human–avatar motor interactions task. Participants were instructed to grasp the object as synchronously as possible with their avatar. They performed opposite (complementary) or same (imitative) movements in different blocks with respect to those of the avatar. The participants did not know in advance where to grasp the bottle, thus they needed to predict the action of the avatar and to adapt to them. (Figure modified from Era *et al.* [41]).

conspecific. Such task recruits analogous processes as human–human interaction, namely mutual adjustment and automatic imitation [39,40]. Importantly, neither the interaction goal nor the participants' own action goal could be achieved without predicting and monitoring the avatar's movements and consequently adapting to them.

Participants seated at a table with a bottle-shaped object placed 45 cm in front of them. An avatar facing the participant was displayed on a monitor, positioned behind the bottle-shaped object. In front of the avatar, there was a virtual object, identical to the one positioned in front of the participant. Participants were asked to reach and grasp the bottle-shaped object placed in front of them. They had their right-hand index-thumb fingers resting over a start button placed 40 cm from the bottle-shaped object and 10 cm to the right of the table's midline. Two pairs of touch-sensitive markers were placed at 15 cm and 23 cm of the object's height and allowed to record the moment in which participants touched the bottle. Because of the shape of the bottle, the virtual character either grasped its lower part with a whole-hand grasp (power grip), or its upper part with a thumb-index finger precision grip. Participants were required to perform opposite (complementary) or same (imitative) movements in different blocks with respect to those of the avatar (which was believed to lead the interaction) without knowing in advance whether the avatar would perform a precision or power grip (see below). In the imitative condition, participants were asked to grasp the same part of the object with the same grasp type as the avatar. In the complementary condition, conversely, participants had to perform the opposite movement as the avatar. The main requirement to perform the task was that participants were instructed to grasp the object as synchronously as possible with their avatar (figure 2).

SMART-D motion capture system (Bioengineering Technology & Systems (B|T|S)) with allowed recording movement kinematics by means of four infrared cameras with wide-angle lenses (sampling rate 100 Hz) placed about 100 cm away from each of the table's four corners. The four cameras captured the movement of reflective infrared markers (5 mm diameter) attached to participants' right upper limbs at the following points: (i) thumb, ulnar side of the nail, (ii) index finger, radial side of the nail, and (iii) wrist, dorso-distal aspect of the radial styloid process.

The intertrial interval was not fixed, but dependent on the time participants took to go back from the bottle to the starting position. The experimenter was manually moving to the next trial as soon as participants went back to the starting position and pressed the start button.

## 5.3. Procedure

The experiment was composed of different phases. The first one was a familiarization phase: participants were asked to perform a human–avatar motor interaction task. They faced an avatar on the screen and were asked to predict its actions in order to plan their own actions towards a bottle-shaped object. More specifically, participants were not directly informed about which part of the bottle-shaped object they had to grasp (either the upper part with a precision grip or the lower part with a power grip) but they were asked to perform either imitative (both performing a precision grip or a power grip) or complementary actions (one performing a precision grip and the other a power grip, or vice versa) with respect to the avatar's ones. Afterwards, they underwent a familiarization phase with the concept guessing task: they were asked to observe five photos and to guess what ACs or CCs each picture evoked, by taking advantage of hints provided by the two different confederates (one for the ACs, one for the CCs). The association between the confederate and ACs/CCs, and the order of administration of the blocks in the familiarization phase were maintained in the real experiment.

The second phase was the experimental one: participants first performed the concept guessing task with either the confederate associated to ACs and CCs by using a subset of the available images (five images), and immediately after they performed the human–avatar motor interaction task. At the end of the concept guessing task and before performing the human–avatar motor interaction task, participants were asked to rate using a VAS scale to what extent they needed others in order to guess the concept associated with the blocks of abstract/concrete pictures. The VAS ranged from 0 to 100, with the extremes corresponding to 'not at all'/' extremely'.

Participants were then asked to perform the interactive task with the avatar. Crucially, the experimenter manipulated their beliefs about the avatar identity by specifying that the virtual agent was controlled by the confederate. In a counterbalanced order, participants believed to interact with an avatar reflecting the movement of the confederate which was helping in guessing ACs or CCs.

Finally, they completed the concept guessing task (last 10 images). Although the perceptual appearance of the avatar remained invariant, participants were told that the avatar was embodying in real time the movements of the corresponding confederates, which were interacting with the participants from another room.

Each interactive task block (imitative, complementary) was composed by 48 trials. Imitative and complementary blocks were counterbalanced in the order. Finally, the identity of the confederate helping with ACs and CCs was counterbalanced across participants. The experimental design was a within-subject design.

# 6. Data analysis

## 6.1. Concept guessing task

The independent variable of the experimental design for the concept guessing task was the Type of Concept (Concrete versus Abstract).

We investigated the following dependent variables:

— Objective helping index: We computed a helping index by dividing the average number of suggestions requested for ACs and CCs by each participant, for their guessing accuracy, respectively, for concrete and abstract blocks. ACs objective helping index = ACs (NSugg/Acc); CCs objective helping index = CCs (NSugg/Acc).
— Subjective helping index: to what extent they think to need the other's help in order to identify CCs/ACs associated with an image measured through a VAS rating scale.

These variables were analysed by means of two-tailed paired sample $t$-tests.

## 6.2. Human–avatar motor interaction task

The independent variables of the experimental design for the human–avatar motor interaction task were: Type of Concept (Concrete versus Abstract); Interaction Type (Imitative versus Complementary) and Movement Type (Precision Grip versus Power Grip).

We excluded from the analysis the trials in which participants: (a) missed the touch-sensitive markers, preventing a response from being recorded, (b) released the start button before the 'go' instruction, or (c) did not respect their Imitative/Complementary instructions. The mean percentage of excluded trials in all the measured variables was mean = 0.121, s.d. = 0.091.

We investigated the following dependent variables:

— accuracy, i.e. number of movements executed correctly (according to the instructions);
— reaction times (RTs), i.e. time from the go-signal to the release of the start button;
— movement times (MTs), i.e. time interval between participants releasing the start button and their index-thumb touching the bottle;
— GAsynchr, i.e. absolute value of time delay between the participants' index-thumb contact times on the bottle-shaped object;
— kinematic indexes: maximum wrist height (MaxH) (mm) describing the wrist trajectory of participants (reaching component of the movement) and indexed by the maximum peak of wrist height on the vertical plane from the level of the table; maximum grip aperture (MaxAp) (mm) the maximum peak of index-thumb three-dimensional Euclidean distance.

Within the single participant, behavioural or kinematic trials that fell 2.5 s.d. above or below each individual mean for each experimental condition were excluded as outliers. The mean percentage of excluded trials was mean = 0.01, s.d. = 0.00 in the GAsynchr variable, mean = 0.02, s.d. = 0.01 in the RTs variable, mean = 0.01, s.d. = 0.01 in the MTs, mean = 0.00, s.d. = 0.00 in the MaxH and mean = 0.01, s.d. = 0.00 in the MaxAp.

Participants with an individual mean 2.5 s.d. above or below the group mean in the GAsynchr variable were excluded from the analyses. This criterion led to the exclusion of one participant. Moreover, four participants were excluded from the MaxAp and two participants from the MaxH analyses because of technical problems during the registration.

We ran linear mixed models (or mixed-effects models, [42], in order to analyse RTs, MTs, GAsynchr and the kinematics measures.

We performed multivariate mixed models with R Studio software (R packages lme4, lsmeans, lmerTest, ggplot2, ggthemes, afex, nlme, mumin v. 3.6.3), having as fixed effects the categorical predictors Type of Concept (Abstract, Concrete), the Interaction Type (Imitative, Complementary), the Movement Type (Precision, Power Grip) and as random intercept the participants.

Statistical significance of fixed effects was determined using type III ANOVA test (the *p*-values for the fixed effects were calculated from an *F*-test on Sattethwaite's approximation), with the mixed function from afex package. *Post hoc* comparisons were performed with the 'Estimated Marginal Means' R package (v. 1.3.3, [43]) via the emmeans function and Tukey correction for multiple comparisons.

In order to investigate a possible relation between the different behavioural variables in the human–avatar motor interaction task, we also ran correlation analyses between the mean of each participants in RTs, MTs and GAsynchr.

A Friedman ANOVA was used to analyse accuracy. In order to directly test the influence of participants' subjective and objective need of other's help on the ability to interact, we ran a second analysis on RTs, MTs, GAsynchr and kinematics measures. The analysis included as continuous predictors two indexes measuring subjective (Subjective need of other's help index) and objective (Objective need of other's help index) participant's need of the other's help when guessing ACs compared with CCs and as categorical predictors the Type of Concept (Abstract versus Concrete), the Interaction Type (Imitative versus Complementary) and the Movement Type (Precision versus Power Grip). See electronic supplementary material for further details on the analyses and results.

In keeping with our hypothesis, we only report significant main effects and interactions involving the Type of Concept (Abstract versus Concrete) as a predictor. All the other main effects and interactions are reported in table 2.

We predicted (a) that the number of suggestions required by participants in the Concept guessing task would be higher with ACs than with CCs, (b) that participants would evaluate that the contribution of others is more important for ACs than for CCs processing, and (c) that the GAsynchr, i.e. the absolute value of time delay between the participant's and avatar index-thumb contact times on the bottle-shaped object would be smaller with AC than CCs' avatar. Finally, (d) we expected that visuo-motor interference would be higher in the ACs condition in comparison with the CCs one.

**Table 2.** All the results of the analyses of the different dependent variables of the human–avatar motor interaction task.

| effects | numDf | denDf | $F$ | Pr(>F) |
|---|---|---|---|---|
| **GAsynchr** | | | | |
| interaction type | 1 | 3703.2 | 26.80 | <0.001 |
| movement type | 1 | 3703.2 | 0.66 | 0.42 |
| type of concept | 1 | 3703.2 | 8.35 | 0.001 |
| interaction type: movement type | 1 | 3703.2 | 507.57 | <0.001 |
| interaction type: type of concept | 1 | 3703.1 | 0.49 | 0.48 |
| movement type: type of concept | 1 | 3703.1 | 0.01 | 0.91 |
| interaction type: movementtype: type of concept | 1 | 3703 | 4.49 | 0.03 |
| **RTs** | | | | |
| interaction type | 1 | 3671 | 11.90 | <0.001 |
| movement type | 1 | 3671 | 0.36 | 0.55 |
| type of concept | 1 | 3671 | 13.12 | <0.001 |
| interaction type: movement type | 1 | 3671 | 3.82 | 0.05 |
| interaction type: type of concept | 1 | 3671 | 0.19 | 0.66 |
| movement type: type of concept | 1 | 3671 | 2.37 | 0.12 |
| interaction type: movement type: type of concept | 1 | 3671 | 1.52 | 0.22 |
| **MTs** | | | | |
| interaction type | 1 | 3705 | 23.54 | <0.001 |
| movement type | 1 | 3705.1 | 13.29 | <0.001 |
| type of concept | 1 | 3705 | 15.05 | <0.001 |
| interaction type: movement type | 1 | 3705 | 16.39 | <0.001 |
| interaction type: type of concept | 1 | 3705 | 0.43 | 0.51 |
| movement type: type of concept | 1 | 3705 | 0.06 | 0.80 |
| interaction type: type of movement: type of concept | 1 | 3705 | 0.70 | 0.40 |
| **max H** | | | | |
| interaction type | 1 | 3399.1 | 0.09 | 0.76 |
| movement type | 1 | 3399.1 | 4479.77 | <0.001 |
| type of concept | 1 | 3399 | 5.85 | 0.02 |
| interaction type: movement type | 1 | 3399.1 | 14.37 | <0.001 |
| interaction type: type of concept | 1 | 3399 | 3.88 | 0.05 |
| movement type: type of concept | 1 | 3399 | 0.29 | 0.59 |
| interaction type: movement type: type of concept | 1 | 3399 | 0.06 | 0.80 |
| **max Ap** | | | | |
| interaction type | 1 | 3045 | 2.52 | 0.11 |
| movement type | 1 | 3045 | 2710.32 | <0.001 |
| type of concept | 1 | 3045 | 6.06 | 0.01 |
| interaction type: movement type | 1 | 3045 | 0.95 | 0.33 |
| interaction type: type of concept | 1 | 3045 | 8.70 | <0.01 |
| movement type: type of concept | 1 | 3045 | 0.05 | 0.82 |
| interaction type: movement type: type of concept | 1 | 3045 | 0.06 | 0.80 |

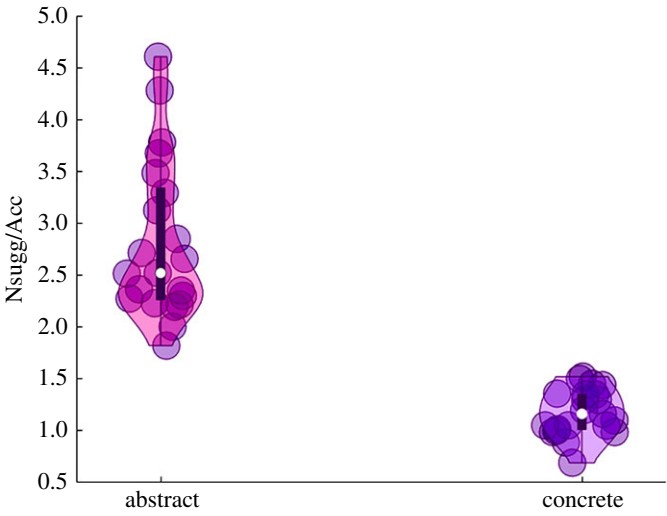

**Figure 3.** Results of the Objective helping index. Paired sample *T*-test on the Objective helping index showed a significant difference between ACs and CCs (*t*(20) = 10.24, *p* < 0.001). Participants showed a higher value of the Objective helping index, which is the ratio between the averaged category suggestions required and the averaged category accuracy. Violin plots display box plots, data density and single subjects' values (dots).

# 7. Results

## 7.1. Concept guessing task

### 7.1.1. Objective helping index

The results of the Objective helping index showed a significant difference between Abstract (mean = 2.82, s.d. = 0.76) and Concrete (mean = 1.18, s.d. = 0.22) concepts (*t*(20) = 10.24, *p* < 0.001, Hedges' g = 2.88). In line with our hypothesis (a) participants showed a higher value of the Objective helping index for ACs compared with CCs, which is the ratio between the averaged suggestions required and the averaged accuracy (figure 3).

### 7.1.2. Subjective helping index

The results of the Subjective helping index indicated a significant difference between Abstract (mean = 74.71, s.d. = 15.39) and Concrete (mean = 38.57, s.d. = 9.04) concepts, (*t*(20) = 6.88, *p* < 0.001, Hedges' g = 2.8).

In keeping with hypothesis (b), participants reported needing the other's help more when they were asked to guess ACs compared with CCs (figure 4).

## 7.2. Human–avatar motor interaction task

### 7.2.1. Accuracy

Friedman ANOVA Chi Square (*N* = 21, d.f. = 7) = 3.93 on the accuracy of task performance in the different experimental conditions resulted in being not significantly different across conditions (*p* = 0.78).

### 7.2.2. Grasping asynchrony

The R$^2$c [44] of the model is = 0.3. The analysis yielded a significant main effect of the Type of Concept (Abstract versus Concrete) ($F_{1,3703.2}$ = 8.35, *p* = 0.001) due to the fact that interacting in the ACs block resulted in smaller GAsynchr (mean = 148 ms, s.e. = 9.03 ms) (i.e. better performance) compared with the CCs block (mean = 158 ms, s.e. = 9.03 ms). In other words, participants were more synchronized when interacting with the avatar that was reputed to embody the confederate who was helping them in guessing the ACs (figure 5).

A significant interaction among the Type of Concept (Abstract versus Concrete), the Interaction Type (Imitative versus Complementary) and the Movement Type (Precision versus Power Grip) predictors

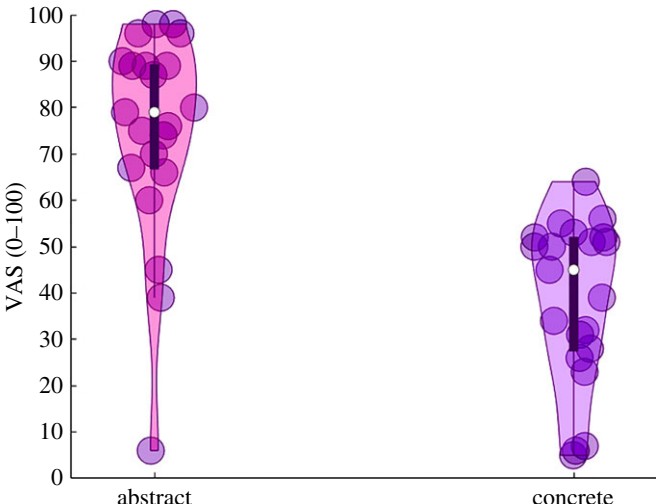

**Figure 4.** Results of the Subjective helping index. Paired sample *T*-test on the Subjective helping index indicated a significant difference between ACs and CCs, ($t(20) = 6.88$, $p < 0.001$). Participants thought to need more the other's help in order to identify the abstract compared with the CCs. Violin plots display box plots, data density and single subjects' values (dots).

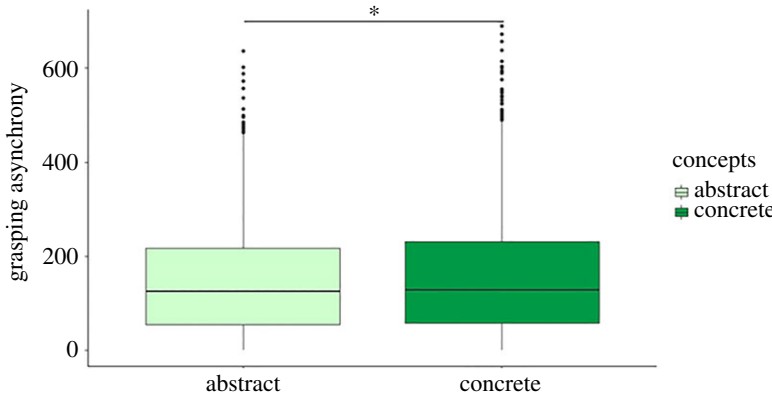

**Figure 5.** Results of the GAsynchr. The analysis on GAsynchr yielded a significant main effect of the Type of Concept (Abstract versus Concrete), ($F_{1,3703.2} = 8.35$, $p = 0.001$) due to the fact that interacting in the ACs block resulted in smaller GAsynchr (better performance) compared with the CCs block. Horizontal lines in the boxes indicate the median, upper and lower borders indicate first and third quartiles; 'whiskers' extend to the farthest points that are not outliers; dots represent outlier trials.

($F_{1,3703} = 4.49$, $p = 0.034$) was found; however, Tukey *post hoc* comparisons resulted to be not significant (all $ps > 0.09$). For all the other main effects and interactions, please see table 2. In line with our hypothesis (c), participants were more synchronous with the ACs versus CCs avatar.

### 7.2.3. Reaction times

The $R^2c$ [44] of the model is = 0.54. The analysis yielded a significant main effect of the Type of Concept (Abstract versus Concrete) ($F_{1,3671} = 13.12$, $p < 0.001$) due to the fact that when interacting in the ACs block participants showed faster RTs (mean = 560 ms, s.e. = 51.3 ms) compared with the CCs block (mean = 586 ms, s.e. = 51.3 ms). Participants started significantly earlier their movements when interacting with the ACs versus CCs avatar. For all the other main effects and interactions, please see table 2.

The correlation between the GAsynchr and the RTs was not significant ($\rho = -0.098$, $p = 0.67$).

### 7.2.4. Movement times

The $R^2c$ [44] of the model is = 0.4. The analysis yielded a significant main effect of the Type of Concept (Abstract versus Concrete) ($F_{1,3705} = 15.05$, $p < 0.001$) due to the fact that when interacting in the ACs block

participants showed longer MTs (mean = 1513 ms, s.e. = 55.2 ms) compared with the CCs block (mean = 1473 ms, s.e. = 55.2 ms). Participants prolonged their movements significantly when interacting with the ACs versus CCs. For all the other main effects and interactions, please see table 2.

The correlation between the MTs and the GAsynchr was not significant ($\rho = -0.142$, $p = 0.53$). However, the correlation between the MTs and the RTs resulted significant ($\rho = -0.932$, $p < 0.001$), suggesting that the MT was linked with the time at which participants were starting the movement, i.e. the earlier they started their movements, the longer the MTs were.

### 7.2.5. Maximum wrist height

The $R^2c$ [44] of the model is = 0.64. The analysis yielded a significant main effect of the Type of Concept (Abstract versus Concrete) ($F_{1,3399} = 5.85$, $p = 0.016$) due to the fact that when interacting in the ACs block participants showed lower MaxH (mean = 197, s.e. = 2.56) compared with the CCs block (mean = 199, s.e. = 2.56). Moreover, the analysis showed a significant two-way Type of Concept × Interaction Type interaction ($F_{1,3399} = 3.88$, $p = 0.049$). Tukey *post hoc* comparisons indicated that when performing complementary movements, MaxH was lower in the abstract (mean = 197 mm, s.e. = 2.59 mm) than in the concrete (mean = 199 mm, s.e. = 2.59 mm) concepts block ($p = 0.016$). The analysis also showed a significant two-way significant Interaction Type × Movement Type interaction ($F_{1,3399} = 14.37$, $p < 0.001$). Tukey *post hoc* comparisons showed that, when grasping the lower part of the bottle, MaxH was higher during Complementary compared with Imitative interactions. This second result revealed visuo-motor interference between executed and observed movements [33–35,41]. Differently from our hypothesis (d), visuo-motor interference was not modulated by the Type of Concept. For all the other main effects and interactions, please see table 2.

### 7.2.6. Maximum grip aperture

The $R^2c$ [44] of the model is = 0.69. The analysis yielded a significant main effect of the Type of Concept (Abstract, Concrete) ($F_{1,3045} = 6.06$, $p = 0.014$) due to the fact that when interacting in the ACs block participants showed smaller MaxAp (mean = 115 mm, s.e. = 2.73 mm) compared with the CCs block (mean = 116 mm, s.e. = 2.73 mm). Moreover, the analysis showed a significant two-way Type of Concept × Interaction Type interaction ($F_{1,3045} = 8.7$, $p = 0.001$). Tukey *post hoc* comparisons indicated that when performing complementary movements, MaxAp was smaller in the abstract (mean = 114 mm, s.e. = 2.74 mm) compared with the concrete (mean = 116 mm, s.e. = 2.74 mm) concepts block ($p < 0.001$). For all the other main effects and interactions, please see table 2.

## 8. Discussion

While recent research has focused on the grounding of ACs in sensorimotor, interoceptive, emotional and linguistic experience, little work has directly investigated the relationship between ACs and sociality. To address this issue, in this study, participants were asked to guess ACs and CCs, having the possibility to be helped by two confederates with whom they were then asked to perform motor interactions. We investigated whether the higher need to be helped in guessing ACs compared with CCs influenced participants' interaction with the confederate in a human–avatar interaction task. Results are straightforward and show that participants performed better when interacting with the avatar who embodied the confederate helping them with ACs compared with the one embodying the confederate helping with CCs.

### 8.1. Other's help is more needed when guessing abstract compared with concrete concepts

We found that participants needed more partner's hints and suggestions when the concepts were abstract rather than concrete (as measured by the Objective helping index). Importantly, participants were metacognitively aware of their higher need of others' help to guess the meaning of ACs than of CCs (as the evaluations on the Subjective helping index indicated). This evidence supports the proposal that, with ACs, we are aware of the limits of our knowledge [29], and that this induces us to prepare ourselves to ask information to others [9,12,24].

## 8.2. Depending on other's help when guessing abstract concepts improves dyadic motor interactions

Using a human–avatar motor interaction task, we found that the need to rely on others influences participants' ability to interact. Participants' performance was more synchronous with the avatar embodying the confederate which they associated with guessing ACs than with that associated with concrete ones. In addition, RTs and MTs analyses revealed that participants started to move earlier, and their MTs were longer with the avatar embodying the confederate who was helping them in guessing ACs rather than CCs. Thus, participants' ability to predict the other's action and integrate it with their own action varied depending on their dependency on other's help. This is in line with the notion that individuals' ability to interact is influenced by the quality of the social relationship among interacting agents [30,35]. The fact that participants interacted more promptly and accurately with the ACs avatar is in line with kinematics studies showing a higher accuracy when interacting with people we take care of [45,46].

Overall, we suggest that the higher synchronism and the prompter performance with the avatar embodying the confederate who provided them hints with ACs are aimed at fostering the collaboration with the other person because participants are aware that they will need more his/her help.

## 8.3. Conceptual processing and social interaction

This work contributes to the literature on conceptual processing at different levels. It investigates the role of the social dimension during the use of ACs compared with CCs, and it makes use of a dynamic, interactive situation.

First, the results have implications for theories of abstractness since they show for the first time and with an interactive paradigm that sociality plays a major role during ACs processing [9]. It has been recently proposed that a higher cohesion level characterizes groups that define their beliefs in terms of abstract ideas [47]. It is possible that the social metacognition mechanism we identified contributes to increasing cohesion between people [48], and the evidence we obtained supports this view.

Second, our study introduces a new method to investigate concepts. As recently underlined by [49], research on ACs has mainly focused on single concepts, in decontextualized situations, while they should be investigated in contexts of 'situated actions'. To the best of our knowledge, the studies adopting interactive paradigms in investigating ACs are only a few. One of the exceptions is the recent experiment [50] in which the authors asked participants to perform an interactive task where they had to explain the meaning of a word to a partner, avoiding mentioning the word (taboo game). In the present study, we adopted an interactive paradigm, in which participants were required first to guess the concepts to which pictures referred and then had to perform a joint task with an avatar embodying another person. This method was able to capture how conceptual use might influence interpersonal coordination in a joint actions. It can thus provide a bridge between the literature on categorization, language and work on joint action [51–53].

The present study demonstrated that when guessing ACs compared with CCs, participants rely more on other people. Which reason subtends this phenomenon? One could object that the effect we found is driven by ACs being more difficult than concrete ones to guess/learn. Because they are more difficult, we would need to rely more on other people; hence we would be more collaborative with others.

First of all, ACs are more difficult to form because their examples are more heterogeneous than those of CCs. Furthermore, ACs are more difficult to process and recall, as revealed by the well-known concreteness effect, i.e. the advantage of concrete words in processing and recognition [54]. Finally, ACs are also explicitly perceived by participants as more difficult overall. In a recent rating study, we asked participants to simply evaluate the 'difficulty' of the written words on a 7-point scale; participants were assigned to different interfering conditions [55]. The conditions influenced the ratings, but across the conditions, ACs were always considered more difficult than concrete ones. The notion of difficulty accounts for the particularity of ACs: ACs are difficult because they are more detached from sensorimotor experience than CCs [48], even if still grounded in sensorimotor and affective properties [11]. Furthermore, they are acquired later, mainly through the linguistic modality [5–7,12,23]. Consequently, they are more flexible [56] and refer to a multitude of contexts (contextual availability [57]), which means that many situations and sensations can represent them. Furthermore, ACs are more heterogeneous since their members have less common features (low-dimensionality: [58,59] and are therefore more variable across and within individuals and cultures.

Moreover, as we have demonstrated in additional validation experiments with the present experiment's stimuli and new ones (see electronic supplementary materials, stimuli validity check), ACs are less associated with images than concrete ones. We think that the consistent pattern of results obtained in the stimuli validity check strongly suggests that the lower association with images is an intrinsic property of ACs compared with CCs.

Notably, a recent study [60] arrives at similar conclusions, showing that facilitation of related over unrelated picture-word combinations is stronger with concrete than with abstract stimuli. Crucially, abstractness and imageability are highly correlated, and for many years, they have been treated as equivalent constructs [61]. Even if not equivalent, the more the concepts grow in abstractness, the less they are imaginable.

In sum: we think we have shown that, because of their difficulty, ACs elicit more pro-social behaviours than CCs. This important objection, the fact that the result mainly depends on difficulty, might, however, lead to very fruitful research. Does the effect extend beyond the guessing task? We have good reasons to believe that it does and that it involves more generally the use of ACs for the aforementioned reasons. Does the effect we found with ACs also extend to other difficult concepts and situations? Do we tend to be more collaborative with others when faced with complex problems that others can help us solve? Further research is needed to address these questions.

Moreover, future studies should investigate whether the increase of synchrony and the improvement in the ability to interact highlighted in the present study is more pronounced with specific categories of ACs, for example, the more difficult or more emotionally connoted ones; whether it occurs only with ACs or whether it also extends to difficult concepts that are not necessarily abstract, for example, scientific ones like 'atom'. This study paves the way to the adoption of more ecological approaches aiming at testing whether the role of social interactions during ACs processing extends to other situations in which abstract words are used, for example, in spontaneous daily conversations that do not involve a guessing task.

Overall, the present study suggests that one of the capacities considered as a hallmark of human cognition, the mastering of ACs, is profoundly grounded in the social dimension.

Ethics. The study was in accordance with the Declaration of Helsinki and approved by the ethical committee of Sapienza University of Rome, Department of Dynamic and Clinical Psychology.

Data accessibility. The experiment was formally preregistered within the OSF at: https://osf.io/4tbme and all data are available at: https://osf.io/98q2g.

Authors' contributions. C.F., V.E., M.C. and A.M.B. developed the study concept and the study design. Testing and data collection were performed by C.F., V.E. and F.D.R. C.F., V.E. and F.D.R. performed the data analysis and interpretation under the supervision of M.C. and A.M.B. C.F., V.E. and A.M.B. drafted the manuscript, and F.D.R. and M.C. provided critical revisions. All authors approved the final version of the manuscript for submission.

Competing Interests. We declare we have no competing interests.

Funding. F.D.R. was supported by INTENSS H2020-MSCA-IF-2017 grant no. 796135; V.E. was supported by the Fondazione Umberto Veronesi and Sapienza Progetti di Ricerca H2020 'Sharetrain' (2018); M.C. was supported by Italian Ministry of Health (RicercaFinalizzata, Giovani Ricercatori 2016, Prot. GR-2016-02361008) and Sapienza University (Progetti di Ricerca Grandi 2020, no. RG120172B8343252); A.M.B. was supported by Sapienza Progetti di Ricerca H2020 (2018, 2019) and H2020-TRAINCREASE-From social interaction to ACs and words: toward human centered technology development; CSA, Proposal no. 952324 P.I. A.M.B.

Acknowledgements. We thank Sarah Boukarras for her help with mixed models analyses. We thank Anna Bianco for her drawings in figures 1 and 2. We thank Giovanna Cuomo for helping pre-processing the kinematics data. We thank Quentin Moreau for helping with the violin plots. We thank Gaetano Tieri, Virtual Reality Lab, Unitelma Sapienza for implementing the avatar. We thank Giovanna M. Massari for helping in data collection.

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
