## [Peer Review File · Royal Society Open Science]

Review History

RSOS-201205.R0 (Original submission)

Review form: Reviewer 1 (Markus Kiefer)

Is the manuscript scientifically sound in its present form?

No

Are the interpretations and conclusions justified by the results?

No

Is the language acceptable?

Yes

Do you have any ethical concerns with this paper?

No

Have you any concerns about statistical analyses in this paper?

No

Recommendation?

Major revision is needed (please make suggestions in comments)

Comments to the Author(s)

The authors investigated the role of social interactions for grounding the meaning of abstract concepts. In the study, participants had to perform a motor interaction task with two avatars who embodied two real confederates. Before and after the motor interaction task, the two confederates provided participants with hints in a concept guessing task: one helped in guessing abstract concepts (ACs) and the other concrete ones (CCs). Participants asked more hints with abstract concepts and were more synchronous when interacting with the avatar corresponding to the abstract concept's confederate. The authors conclude that their results highlight an important role of sociality in grounding abstract concepts.

The topic of the study, the grounding of abstract concepts in sociality, is interesting and novel. The experiment is innovative and could contribute to a better understanding of the representation of abstract concepts. Nevertheless, several issues must be addressed in a revision.

- 1.) I missed detailed information about the selection of the abstract concepts used in the study. The meaning of abstract concepts is quite heterogeneous. As it has been shown in previous work, only a subgroup refers to sociality. How can the meaning of the selected concepts be described? I also did not find information about the number of concepts and associated pictures, which were presented in the experiment. Finally and most importantly, how well were pictures associated with ACs and CCs matched with regard to their relation to the assigned abstract concepts. Any differential relations between ACs and CCs could influence performance in the guessing task and the Need of other indices. I would like to see rating and performance data, i.e. performance in a semantic relatedness judgment task, which indicate that semantic relatedness/task difficulty of the selected pictures were matched for ACs and CCs.
- 2.) The authors argue that social interactions associated with guessing the meaning of abstract concepts influenced performance in the subsequent avatar interaction task. However, alternatively it could be argued that the meaning of the selected abstract concepts was related to sociality and therefore primed synchronization in the avatar interaction task. Can the authors rule out this alternative interpretation?
- 3.) The mechanisms underlying the putative carry-over effects of the guessing task to the avatar interaction task should be better explained in the introduction section. This is particular important because the two tasks are not directly related in their conceptual content, besides the presence of the confederates. What is the theoretical justification to expect such carry-over effects? Which role does the helpfulness of the confederate play? I also missed an analysis elucidating the role of the helpfulness of the confederate, which could affect synchrony in the avatar interaction task. I also had problem to link hypothesis c) about automatic imitation with the main aim of the study.
Minor points
- 4.) Were t-tests two-tailed or one-tailed?
- 5.) Were confederates physically present during the guessing tasks or just simulated on the computer?
- 6.) The authors missed to quote in the introduction earlier work proposing a multiple representation theory of abstract concepts, which emphasizes the role of social interactions, in addition to sensory-motor, emotional, introspective and linguistic information (Kiefer, M., &

Harpaintner, M. (2020). Varieties of abstract concepts and their grounding in perception or action. *Open Psychology*, 2, 119-137.). The authors should also acknowledge the heterogeneity of the meaning of abstract concepts as suggested by earlier work (Ghio, M., Vaghi, M. M. S., & Tettamanti, M. (2013). Fine-grained semantic categorization across the abstract and concrete domains. *Plos One*, 8(6); Harpaintner, M., Trumpp, N. M., & Kiefer, M. (2018). The semantic content of abstract concepts: A property listing study of 296 abstract words. *Frontiers in Psychology*, 9, 1748; Hoffman, P. (2016). The meaning of 'life' and other abstract words: Insights from neuropsychology. *Journal of Neuropsychology*, 10(2), 317-343). Finally, the authors should also mention that the importance of visual and motor information for abstract concepts has been demonstrated in a recent fMRI study (Harpaintner, M., Sim, E. J., Trumpp, N. M., Ulrich, M., & Kiefer, M. (2020). The grounding of abstract concepts in the motor and visual system: An fMRI study. *Cortex*, 124, 1-22.).

7.) line 361: "In line with our hypothesis (a), participants asked more suggestions and committed more errors" This statement is not correct because the Objective Helping index, to which this statement refers, is calculated as RATIO between number of requested suggestions and accuracy.

8.) The authors write on p. 18 that the study is preregistered on <https://osf.io/98q2g/>. Unfortunately, I did not find the preregistered study protocol describing the goal of the study, hypotheses, methods and data analyses, but only some data files and one R script including LMM analyses. When clicking in osf on the registrations button, I only receive the message "There have been no completed registrations of this project. Perhaps I missed to find the location of this information on the osf website. I also did not find the data files containing the Objective and subjective Helping indices and the R-scripts for the t-tests and analyses described in the supplement.

9.) line 474: The authors discuss visuo-motor interference effects influenced by subjective need of help. However, the analyses were only reported in the quite lengthy supplement (which might be shortened). If this finding is deemed important, the corresponding analyses should be moved to the main text.

Review form: Reviewer 2

Is the manuscript scientifically sound in its present form?

Yes

Are the interpretations and conclusions justified by the results?

No

Is the language acceptable?

Yes

Do you have any ethical concerns with this paper?

No

Have you any concerns about statistical analyses in this paper?

No

Recommendation?

Major revision is needed (please make suggestions in comments)

Comments to the Author(s)

In this paper, the authors investigate the relationship between concept abstract/concreteness and sociality, by measuring the degree to which participants coordinate their actions with confederates. One confederate gave hints to a concrete concept (CC); the other to an abstract (AC). Coordination was measured in a motor interaction task with avatars corresponding to the confederates. There was more coordination with the AC hinting confederate, suggesting a link between conceptual abstraction and sociality or recruitment of others.

I think that this is an ingenious idea and a well executed experiment and analysis, and I commend the authors. My main comment, which I think needs to be addressed before publication, concerns a confound which I think questions the claims that the wish to make.

My worry is that the difference in behaviour towards AC and CC confederates is being driven by task difficulty alone.

Imagine a task with maths multiple choice questions (or general knowledge questions). There were easy and hard categories (that we can objectively assess, in the way that AC and CC questions had an objective difference in difficulty in this experiment). The two confederates only give hints to the easy or the hard maths/general knowledge questions respectively.

Isn't it very plausible, in this task, that the participant will become more behaviourally coupled to the confederate giving the hints to the hard question. This is because that confederate is seen as smarter, more valuable in this context, more pro social and helpful, etc.

If we did get that result in my imaginary task, it would suggest that such differences in motor coordination are telling us about how people recruit others in the service of more difficult tasks, by recruiting behavioural coupling. That's a really neat finding! But it would mean that the results the authors present here don't tell us anything about concepts and their structure, or the the social nature of ACs vs CCs.

So, to accept the authors current conclusions, I would want to see data from a task like this, mashed as closely as possible to the conceptual one, where there were similarly two levels of difficulty, but with no relation to concept concreteness. If behavioural coordination was modulated exclusively in the conceptual task, then I would buy the conclusion that it is the social content of ASc specifically that is the reason they modulate motor coupling.

Decision letter (RSOS-201205.R0)

Dear Dr Fini,

The Editors assigned to your paper RSOS-201205 "Abstract concepts in interaction: The need of others when guessing abstract concepts smooths dyadic motor interactions" have now received comments from reviewers and would like you to revise the paper in accordance with the reviewer comments and any comments from the Editors. Please note this decision does not guarantee eventual acceptance.

Please submit your revised manuscript and required files (see below) no later than 21 days from today's (ie 07-Sep-2020) date. Note: the ScholarOne system will 'lock' if submission of the revision is attempted 21 or more days after the deadline. If you do not think you will be able to meet this deadline please contact the editorial office immediately.

on behalf of Dr Rochelle Ackerley (Associate Editor) and Essi Viding (Subject Editor)
openscience@royalsociety.org

Associate Editor Comments to Author (Dr Rochelle Ackerley):

Two reviewers have assessed your manuscript and found it to be novel and interesting. However, both have raised major questions about the work, especially the theoretical implications, which need to be fully addressed. In addition, the authors need to be clear about the preregistration of their work, as they actually provide two links in their manuscript – one to the preregistered document and another to data. It would also be good if the authors could comment in their paper as to whether their submitted work deviated from the preregistered plan.

Reviewer comments to Author:

Reviewer: 1

Comments to the Author(s)

The authors investigated the role of social interactions for grounding the meaning of abstract concepts. In the study, participants had to perform a motor interaction task with two avatars who embodied two real confederates. Before and after the motor interaction task, the two confederates provided participants with hints in a concept guessing task: one helped in guessing abstract concepts (ACs) and the other concrete ones (CCs). Participants asked more hints with abstract concepts and were more synchronous when interacting with the avatar corresponding to the

abstract concept's confederate. The authors conclude that their results highlight an important role of sociality in grounding abstract concepts.

The topic of the study, the grounding of abstract concepts in sociality, is interesting and novel. The experiment is innovative and could contribute to a better understanding of the representation of abstract concepts. Nevertheless, several issues must be addressed in a revision.

1.) I missed detailed information about the selection of the abstract concepts used in the study. The meaning of abstract concepts is quite heterogeneous. As it has been shown in previous work, only a subgroup refers to sociality. How can the meaning of the selected concepts be described? I also did not find information about the number of concepts and associated pictures, which were presented in the experiment. Finally and most importantly, how well were pictures associated with ACs and CCs matched with regard to their relation to the assigned abstract concepts. Any differential relations between ACs and CCs could influence performance in the guessing task and the Need of other indices. I would like to see rating and performance data, i.e. performance in a semantic relatedness judgment task, which indicate that semantic relatedness/task difficulty of the selected pictures were matched for ACs and CCs.

2.) The authors argue that social interactions associated with guessing the meaning of abstract concepts influenced performance in the subsequent avatar interaction task. However, alternatively it could be argued that the meaning of the selected abstract concepts was related to sociality and therefore primed synchronization in the avatar interaction task. Can the authors rule out this alternative interpretation?

3.) The mechanisms underlying the putative carry-over effects of the guessing task to the avatar interaction task should be better explained in the introduction section. This is particular important because the two tasks are not directly related in their conceptual content, besides the presence of the confederates. What is the theoretical justification to expect such carry-over effects? Which role does the helpfulness of the confederate play? I also missed an analysis elucidating the role of the helpfulness of the confederate, which could affect synchrony in the avatar interaction task. I also had problem to link hypothesis c) about automatic imitation with the main aim of the study.

Minor points

4.) Were t-tests two-tailed or one-tailed?

5.) Were confederates physically present during the guessing tasks or just simulated on the computer?

6.) The authors missed to quote in the introduction earlier work proposing a multiple representation theory of abstract concepts, which emphasizes the role of social interactions, in addition to sensory-motor, emotional, introspective and linguistic information (Kiefer, M., & Harpaintner, M. (2020). Varieties of abstract concepts and their grounding in perception or action. *Open Psychology*, 2, 119-137.). The authors should also acknowledge the heterogeneity of the meaning of abstract concepts as suggested by earlier work (Ghio, M., Vaghi, M. M. S., & Tettamanti, M. (2013). Fine-grained semantic categorization across the abstract and concrete domains. *Plos One*, 8(6); Harpaintner, M., Trumpp, N. M., & Kiefer, M. (2018). The semantic content of abstract concepts: A property listing study of 296 abstract words. *Frontiers in Psychology*, 9, 1748; Hoffman, P. (2016). The meaning of 'life' and other abstract words: Insights from neuropsychology. *Journal of Neuropsychology*, 10(2), 317-343). Finally, the authors should also mention that the importance of visual and motor information for abstract concepts has been demonstrated in a recent fMRI study (Harpaintner, M., Sim, E. J., Trumpp, N. M., Ulrich, M., &

Kiefer, M. (2020). The grounding of abstract concepts in the motor and visual system: An fMRI study. *Cortex*, 124, 1-22.).

7.) line 361: "In line with our hypothesis (a), participants asked more suggestions and committed more errors" This statement is not correct because the Objective Helping index, to which this statement refers, is calculated as RATIO between number of requested suggestions and accuracy.

8.) The authors write on p. 18 that the study is preregistered on <https://osf.io/98q2g/>. Unfortunately, I did not find the preregistered study protocol describing the goal of the study, hypotheses, methods and data analyses, but only some data files and one R script including LMM analyses. When clicking in osf on the registrations button, I only receive the message "There have been no completed registrations of this project. Perhaps I missed to find the location of this information on the osf website. I also did not find the data files containing the Objective and subjective Helping indices and the R-scripts for the t-tests and analyses described in the supplement.

9.) line 474: The authors discuss visuo-motor interference effects influenced by subjective need of help. However, the analyses were only reported in the quite lengthy supplement (which might be shortened). If this finding is deemed important, the corresponding analyses should be moved to the main text.

Reviewer: 2

Comments to the Author(s)

In this paper, the authors investigate the relationship between concept abstract/concreteness and sociality, by measuring the degree to which participants coordinate their actions with confederates. One confederate gave hints to a concrete concept (CC); the other to an abstract (AC). Coordination was measured in a motor interaction task with avatars corresponding to the confederates. There was more coordination with the AC hinting confederate, suggesting a link between conceptual abstraction and sociality or recruitment of others.

I think that this is an ingenious idea and a well executed experiment and analysis, and I commend the authors. My main comment, which I think needs to be addressed before publication, concerns a confound which I think questions the claims that the wish to make.

My worry is that the difference in behaviour towards AC and CC confederates is being driven by task difficulty alone.

Imagine a task with maths multiple choice questions (or general knowledge questions). There were easy and hard categories (that we can objectively assess, in the way that AC and CC questions had an objective difference in difficulty in this experiment). The two confederates only give hints to the easy or the hard maths/general knowledge questions respectively.

Isn't it very plausible, in this task, that the participant will become more behaviourally coupled to the confederate giving the hints to the hard question. This is because that confederate is seen as smarter, more valuable in this context, more pro social and helpful, etc.

If we did get that result in my imaginary task, it would suggest that such differences in motor coordination are telling us about how people recruit others in the service of more difficult tasks, by recruiting behavioural coupling. That's a really neat finding! But it would mean that the results the authors present here don't tell us anything about concepts and their structure, or the social nature of ACs vs CCs.

So, to accept the authors current conclusions, I would want to see data from a task like this, mashed as closely as possible to the conceptual one, where there were similarly two levels of difficulty, but with no relation to concept concreteness. If behavioural coordination was modulated exclusively in the conceptual task, then I would buy the conclusion that it is the social content of ASc specifically that is the reason they modulate motor coupling.

===PREPARING YOUR MANUSCRIPT===

===PREPARING YOUR REVISION IN SCHOLARONE===

Author's Response to Decision Letter for (RSOS-201205.R0)

See Appendix A.

RSOS-201205.R1 (Revision)

Review form: Reviewer 1 (Markus Kiefer)

Is the manuscript scientifically sound in its present form?

No

Are the interpretations and conclusions justified by the results?

No

Is the language acceptable?

Yes

Do you have any ethical concerns with this paper?

No

Have you any concerns about statistical analyses in this paper?

No

Recommendation?

Major revision is needed (please make suggestions in comments)

Comments to the Author(s)

This is a revised manuscript, which I have already evaluated previously. Overall, the authors have improved the manuscript in several respects, as suggested by the reviewers. However, the authors were not responsive with regard to some relevant aspects. Several concerns therefore remain and require further improvements

- 1.) I still would like to see rating and performance data, i.e. performance in a semantic relatedness judgment task, which indicate that semantic relatedness/task difficulty of the selected pictures were matched for ACs and CCs. The new control study using a guessing task is not sensitive enough to reveal differences in the stimulus pairs between ACs and CCs.
- 2.) The authors now argue in the discussion section (p. 17) that ACs are more difficult, so that individuals would rely more on other people and would be more collaborative with them. If this is a mere effect of difficulty, the same effect should also show up for more difficult CCs. To support their claim, the authors could run an analysis on CCs with conceptual difficulty as additional factor.

Decision letter (RSOS-201205.R1)

Dear Dr Fini

The Editors assigned to your paper RSOS-201205.R1 "Abstract concepts in interaction: The need of others when guessing abstract concepts smooths dyadic motor interactions" have now received

comments from a reviewer and would like you to revise the paper in accordance with the reviewer comments and any comments from the Editors. Please note this decision does not guarantee eventual acceptance.

Please submit your revised manuscript and required files (see below) no later than 21 days from today's (ie 04-Dec-2020) date. Note: the ScholarOne system will 'lock' if submission of the revision is attempted 21 or more days after the deadline. If you do not think you will be able to meet this deadline please contact the editorial office immediately.

on behalf of Dr Rochelle Ackerley (Associate Editor) and Essi Viding (Subject Editor)
openscience@royalsociety.org

Associate Editor Comments to Author (Dr Rochelle Ackerley):

Comments to the Author:

I have read the paper myself and we have received further comments from one reviewer. I agree with both reviewers' initial evaluations that your work on the grounding of abstract concepts in sociality is interesting and novel, and it adds to the literature. Previously, both reviewers had concerns about the work, especially about the level of difficulty. Reviewer 1 also wanted to see the rating and performance data; both they and I feel that this has not been properly addressed in your reply to the comments or in the manuscript. I would like you to further address the issue of difficulty and rating/performance data, as per Reviewer 1's additional comments. I also have a few minor points in the methods which need addressing:

- Did the participants give written informed consent?
- The authors have now added 'The analyses including the covariates have been moved to the supplementary materials instead of being part of the main text'. This phrasing is a little strange

here and it would be far simpler to say something like: The analyses including the covariates can be found in the supplementary materials.

- The text that has been added on p.6 of the methods seems to be in the wrong tense, for example, 'we have conducted'. This can all be corrected by removing 'have' for each occasion in this new text.

Reviewer comments to Author:

Reviewer: 1

Comments to the Author(s)

This is a revised manuscript, which I have already evaluated previously. Overall, the authors have improved the manuscript in several respects, as suggested by the reviewers. However, the authors were not responsive with regard to some relevant aspects. Several concerns therefore remain and require further improvements

1.) I still would like to see rating and performance data, i.e. performance in a semantic relatedness judgment task, which indicate that semantic relatedness/task difficulty of the selected pictures were matched for ACs and CCs. The new control study using a guessing task is not sensitive enough to reveal differences in the stimulus pairs between ACs and CCs.

2.) The authors now argue in the discussion section (p. 17) that ACs are more difficult, so that individuals would rely more on other people and would be more collaborative with them. If this is a mere effect of difficulty, the same effect should also show up for more difficult CCs. To support their claim, the authors could run an analysis on CCs with conceptual difficulty as additional factor.

===PREPARING YOUR MANUSCRIPT===

If you have been asked to revise the written English in your submission as a condition of publication, you must do so, and you are expected to provide evidence that you have received language editing support. The journal would prefer that you use a professional language editing service and provide a certificate of editing, but a signed letter from a colleague who is a native speaker of English is acceptable. Note the journal has arranged a number of discounts for authors

using professional language editing services
(<https://royalsociety.org/journals/authors/benefits/language-editing/>).

===PREPARING YOUR REVISION IN SCHOLARONE===

<https://royalsociety.org/journals/authors/author-guidelines/#supplementary-material> to include a suitable title and informative caption. An example of appropriate titling and captioning may be found at https://figshare.com/articles/Table_S2_from_Is_there_a_trade-

off_between_peak_performance_and_performance_breadth_across_temperatures_for_aerobic_sc
ope_in_teleost_fishes_/3843624.

Author's Response to Decision Letter for (RSOS-201205.R1)

See Appendix B.

RSOS-201205.R2 (Revision)

Review form: Reviewer 1 (Markus Kiefer)

Is the manuscript scientifically sound in its present form?

No

Are the interpretations and conclusions justified by the results?

No

Is the language acceptable?

Yes

Do you have any ethical concerns with this paper?

No

Have you any concerns about statistical analyses in this paper?

Yes

Recommendation?

Major revision is needed (please make suggestions in comments)

Comments to the Author(s)

This is a revised manuscript, which I have already evaluated previously. I would like to thank the authors for their detailed response to my two concerns raised in the previous review round. In particular, it is commendable that they performed additional control experiments including the semantic relatedness judgment task, which I have previously proposed.

The results from these control studies clearly indicate that for ACs pictures are more difficult to match than for CCs. I agree with the authors' claim that this might be an intrinsic property of ACs. However, in the revised manuscript the new control studies are only described and discussed in the Supplementary Material. It is essential for a proper interpretation of the entire findings of the study that the main text including the abstract reflects the results of the new control studies. For that reason, the discussion of these control studies must be moved from the Supplementary Material to the main text, for instance to the discussion section. The abstract

should also reflect this balanced interpretation of the results in terms of conceptual difficulty. The control studies in the Supplementary Material should also be referred to in the Methods section.

Furthermore, the effect of conceptual difficulty of the CCs and AS on objective and subjective help indices in the guessing task must be assessed by entering RTs to the individual ACs and CCs from the relatedness judgment task as covariate in LMM analyses. It is important to see, whether or not the effect of conceptual category disappears when conceptual difficulty is entered as covariate. If yes, this would slightly alter the interpretation of the results.

Finally and importantly: The author's study is pre-registered on the OSF platform. In order to conform with the Open Science guidelines, it is essential to indicate in the main text, which parts of the methods and analyses deviate from the pre-registration and are therefore exploratory, rather than confirmatory. Therefore, the sentence on p. 5 lines 136-138 must be reworded accordingly because it does not correctly capture all facets of the study: "All the hypotheses, experimental procedures, and data analyses have been specified in a pre-registration <https://osf.io/4tbme>. The analyses including the covariates can be found in the supplementary materials."

Decision letter (RSOS-201205.R2)

Dear Dr Fini

The Editors assigned to your paper RSOS-201205.R2 "Abstract concepts in interaction: The need of others when guessing abstract concepts smooths dyadic motor interactions" have now received comments from reviewers and would like you to revise the paper in accordance with the reviewer comments and any comments from the Editors. Please note this decision does not guarantee eventual acceptance.

Please submit your revised manuscript and required files (see below) no later than 21 days from today's (ie 06-May-2021) date. Note: the ScholarOne system will 'lock' if submission of the revision is attempted 21 or more days after the deadline. If you do not think you will be able to meet this deadline please contact the editorial office immediately.

Please note article processing charges apply to papers accepted for publication in Royal Society Open Science (<https://royalsocietypublishing.org/rsos/charges>). Charges will also apply to papers transferred to the journal from other Royal Society Publishing journals, as well as papers submitted as part of our collaboration with the Royal Society of Chemistry

(<https://royalsocietypublishing.org/rsos/chemistry>). Fee waivers are available but must be requested when you submit your revision (<https://royalsocietypublishing.org/rsos/waivers>).

on behalf of Dr Rochelle Ackerley (Associate Editor) and Essi Viding (Subject Editor)
openscience@royalsociety.org

Associate Editor Comments to Author (Dr Rochelle Ackerley):

Comments to the Author:

Further comments have been received from one reviewer and these should be taken into consideration when modifying your paper. The reviewer makes a very good point about the pre-registration of studies and how these rules need to be integrated into your work.

Reviewer comments to Author:

Reviewer: 1

Comments to the Author(s)

This is a revised manuscript, which I have already evaluated previously. I would like to thank the authors for their detailed response to my two concerns raised in the previous review round. In particular, it is commendable that they performed additional control experiments including the semantic relatedness judgment task, which I have previously proposed.

The results from these control studies clearly indicate that for ACs pictures are more difficult to match than for CCs. I agree with the authors' claim that this might be an intrinsic property of ACs. However, in the revised manuscript the new control studies are only described and discussed in the Supplementary Material. It is essential for a proper interpretation of the entire findings of the study that the main text including the abstract reflects the results of the new control studies. For that reason, the discussion of these control studies must be moved from the Supplementary Material to the main text, for instance to the discussion section. The abstract should also reflect this balanced interpretation of the results in terms of conceptual difficulty. The control studies in the Supplementary Material should also be referred to in the Methods section.

Furthermore, the effect of conceptual difficulty of the CCs and AS on objective and subjective help indices in the guessing task must be assessed by entering RTs to the individual ACs and CCs from the relatedness judgment task as covariate in LMM analyses. It is important to see, whether or not the effect of conceptual category disappears when conceptual difficulty is entered as covariate. If yes, this would slightly alter the interpretation of the results.

Finally and importantly: The author's study is pre-registered on the OSF platform. In order to conform with the Open Science guidelines, it is essential to indicate in the main text, which parts of the methods and analyses deviate from the pre-registration and are therefore exploratory, rather than confirmatory. Therefore, the sentence on p. 5 lines 136-138 must be reworded accordingly because it does not correctly capture all facets of the study: "All the hypotheses, experimental procedures, and data analyses have been specified in a pre-

registration <https://osf.io/4tbme>. The analyses including the covariates can be found in the supplementary materials."

===PREPARING YOUR MANUSCRIPT===

===PREPARING YOUR REVISION IN SCHOLARONE===

- 1) One version identifying all the changes that have been made (for instance, in coloured highlight, in bold text, or tracked changes);
 - 2) A 'clean' version of the new manuscript that incorporates the changes made, but does not highlight them.
 - An individual file of each figure (EPS or print-quality PDF preferred [either format should be produced directly from original creation package], or original software format).
 - An editable file of each table (.doc, .docx, .xls, .xlsx, or .csv).
 - An editable file of all figure and table captions.
- Note: you may upload the figure, table, and caption files in a single Zip folder.
- Any electronic supplementary material (ESM).
 - If you are requesting a discretionary waiver for the article processing charge, the waiver form must be included at this step.
 - If you are providing image files for potential cover images, please upload these at this step, and inform the editorial office you have done so. You must hold the copyright to any image provided.
 - A copy of your point-by-point response to referees and Editors. This will expedite the preparation of your proof.

- Ensure that your data access statement meets the requirements at <https://royalsociety.org/journals/authors/author-guidelines/#data>. You should ensure that you cite the dataset in your reference list. If you have deposited data etc in the Dryad repository, please include both the 'For publication' link and 'For review' link at this stage.
- If you are requesting an article processing charge waiver, you must select the relevant waiver option (if requesting a discretionary waiver, the form should have been uploaded at Step 3 'File upload' above).
- If you have uploaded ESM files, please ensure you follow the guidance at <https://royalsociety.org/journals/authors/author-guidelines/#supplementary-material> to include a suitable title and informative caption. An example of appropriate titling and captioning may be found at https://figshare.com/articles/Table_S2_from_Is_there_a_trade-off_between_peak_performance_and_performance_breadth_across_temperatures_for_aerobic_sc_ope_in_teleost_fishes_/3843624.

Author's Response to Decision Letter for (RSOS-201205.R2)

See Appendix C.

Decision letter (RSOS-201205.R3)

Dear Dr fini,

It is a pleasure to accept your manuscript entitled "Abstract concepts in interaction: The need of others when guessing abstract concepts smooths dyadic motor interactions" in its current form for publication in Royal Society Open Science.

on behalf of Dr Rochelle Ackerley (Associate Editor) and Essi Viding (Subject Editor)
openscience@royalsociety.org

Appendix A

Dear Dr Fini,

The Editors assigned to your paper RSOS-201205 "Abstract concepts in interaction: The need of others when guessing abstract concepts smooths dyadic motor interactions" have now received comments from reviewers and would like you to revise the paper in accordance with the reviewer comments and any comments from the Editors. Please note this decision does not guarantee eventual acceptance.

Please submit your revised manuscript and required files (see below) no later than 21 days from today's (ie 07-Sep-2020) date. Note: the ScholarOne system will 'lock' if submission of the revision is attempted 21 or more days after the deadline. If you do not think you will be able to meet this deadline please contact the editorial office immediately.

Best regards,

on behalf of Dr Rochelle Ackerley (Associate Editor) and Essi Viding (Subject Editor)
openscience@royalsociety.org

Associate Editor Comments to Author (Dr Rochelle Ackerley):

Two reviewers have assessed your manuscript and found it to be novel and interesting. However, both have raised major questions about the work, especially the theoretical implications, which need to be fully addressed. In addition, the authors need to be clear about the preregistration of their

work, as they actually provide two links in their manuscript – one to the preregistered document and another to data. It would also be good if the authors could comment in their paper as to whether their submitted work deviated from the preregistered plan.

Dear Prof. Ackerley,

We are grateful for the opportunity to revise our manuscript and we wish to thank you and the reviewers for the thorough reading of our paper and the very helpful comments and suggestions. We provide detailed answers to each of the comments. In the revised manuscript the changes are all marked in bold.

Our submitted manuscript sticks to the planned work in the preregistered plan; we want only to clarify that the analyses including the covariates have been moved to the supplementary materials instead of being part of the main text. This information has been also specified inside the manuscript (lines 136-137) pag (5).

We hope you agree that we did our best to address all the comments and that our paper improved accordingly. We, therefore, hope that you will find this revised manuscript suitable for publication in Royal Society Open Science.

Sincerely,

Chiara Fini, Vanessa Era, Federico Da Rold, Matteo Candidi, A.M. Borghi

Reviewer comments to Author:

Reviewer: 1

Comments to the Author(s)

The authors investigated the role of social interactions for grounding the meaning of abstract concepts. In the study, participants had to perform a motor interaction task with two avatars who embodied two real confederates. Before and after the motor interaction task, the two confederates provided participants with hints in a concept guessing task: one helped in guessing abstract concepts (ACs) and the other concrete ones (CCs). Participants asked more hints with abstract concepts and were more synchronous when interacting with the avatar corresponding to the abstract concept's confederate. The authors conclude that their results highlight an important role of sociality in grounding abstract concepts.

The topic of the study, the grounding of abstract concepts in sociality, is interesting and novel. The experiment is innovative and could contribute to a better understanding of the representation of abstract concepts. Nevertheless, several issues must be addressed in a revision.

We thank the Reviewer for the positive comments regarding the manuscript. We are grateful for the opportunity to improve the quality of our paper. Please find below the answer to each point raised by the Reviewer.

1) I missed detailed information about the selection of the abstract concepts used in the study. The meaning of abstract concepts is quite heterogeneous. As it has been shown in previous work, only a subgroup refers to sociality. How can the meaning of the selected concepts be described? I also did not find information about the number of concepts and associated pictures, which were presented in the experiment.

- ❖ The Reviewer addresses an important issue. The concrete and abstract concepts have been selected in order to be matched along the dimension of Familiarity, Age of Acquisition (AoA) and Modality of Acquisition (MoA), see Table 1) in the manuscript. The concepts have been selected taking into account the variety of meaning from the dataset of Della Rosa et al. (2010).
- ❖ We have accurately selected the pictures representing the abstract and concrete concepts on the basis of the most frequent contextual definitions (among six) provided by an independent sample of 10 Master students. We gave them each concept name and asked them to write six situations related to it. For each concept, we selected the situation most frequently produced across participants. If two situations obtained the same score, we selected the situation that was produced earlier, assuming that it was more accessible to participants. Furthermore, we now provide the number of concepts and the associated pictures (40: 20 for abstract and 20 for concrete concepts) used in the guessing task. In the *OSF link* <https://osf.io/98q2g>, (folder named “Materials”), we have also entered all the pictures associated with the concepts.

Finally and most importantly, how well were pictures associated with ACs and CCs matched with regard to their relation to the assigned abstract concepts. Any differential relations between ACs and CCs could influence performance in the guessing task and the Need of other indices. I would like to see rating and performance data, i.e. performance in a semantic relatedness judgment task, which indicate that semantic relatedness/task difficulty of the selected pictures were matched for ACs and CCs.

- ❖ We thank the reviewer for raising this issue. The issue of difficulty per se is a complex matter, we have addressed it adding a passage in the general discussion (see also the response to Reviewer 2). As to the semantic relation between the pictures and the concept, we have now explained how we selected the pictures. For both ACs and CCs, we have adopted the same criterion, selecting the image representing the situation that was most frequently associated to the concept.

We have added this information also in the manuscript line (163) pag (6):

- ❖ *“We selected a list of 40 (20 for abstract and 20 for concrete concepts) words from the database of Della Rosa et al. (2010), see Table 1). In order to prepare the material, we asked 10 university students who did not take part in the main experiment to produce 6 situations associated with each word. We then found a corresponding picture for the situation that participants produced more frequently for each word.”*
- ❖ Moreover, in order to demonstrate that the pictures selected for the abstract and the concrete concepts were equally valid, we have conducted an on-line survey involving 26 participants (age= 38.5 st dev=8.92; 21 females). Specifically, participants were asked to write the concept that they thought to be represented by each picture. Then, we have calculated the percent accuracy (how many times the concept was guessed by all the participants) for each abstract and concrete picture. The accuracy score was very low either in the abstract (ACC= .092 st dev=.15) and in the concrete concept’s category (ACC= .034 st dev=.05). The two tailed paired t-test comparison show that there was no statistical difference between the percent accuracy in abstract and concrete concepts ($t(19)=1.62, p=.12$). Thus, the semantic relations between the pictures and the concrete/abstract concepts was not easily detectable; indeed, we voluntarily chose ambiguous, contextual, social scenarios associated with the concepts in order to control either for the imageability

and for their social dimension. In conclusion, the semantic relations between the pictures and the concrete/abstract concepts were matched.

We have now added this information in the manuscript at lines (167-175), pag (6) : *“In order to demonstrate that the pictures selected to represent the abstract and the concrete concepts were equally valid, we have conducted an on-line survey involving 26 participants. Specifically, participants were asked to write the concept that they thought to be represented by each picture. Then, we calculated the percent accuracy (how many times the concept was guessed by participants) for each abstract and concrete picture. Accuracy scores were very low in the abstract concepts (ACC= .092 st dev=.15) and in the concrete concepts (ACC= .034 st dev=.05). The two tailed paired t-test comparisons show that there was no statistical difference between the percent accuracy in abstract and concrete concepts ($t(19)=1.62, p=.12$).”*

2.) The authors argue that social interactions associated with guessing the meaning of abstract concepts influenced performance in the subsequent avatar interaction task. However, alternatively it could be argued that the meaning of the selected abstract concepts was related to sociality and therefore primed synchronization in the avatar interaction task. Can the authors rule out this alternative interpretation?

- ❖ **To control whether the content of abstract concepts is more social than that of concrete ones, we coded the stimuli according to two criteria. First, we coded the selected concepts in order to verify whether they referred to a social situation. The social situations were six for both concrete and abstract concepts, namely: abstract concepts: 1) “friendship”: hugging people, 2) “discussion”: people talking among each other, 3) “justification”: present the justification booklet at school, 4) “injustice”: raise the hands against someone (metaphorical for hitting someone), 5) “education”: scolding; 6) “beginning”= the start of a race; concrete concepts: 1) guest: offer dinner, 2) fleet: full of sailors and crew, 3) weapon: policeman usually have them, 4) army: soldiers who wear uniforms, 5) camping-place: people around the fire 6) telegraph: a girl trying to communicate through it.**

Second, we checked the number of people presented in the images that participants received. We first considered the images in which more than one person was present. In the case of abstract concepts, 6 images included more than one person, while in the case of concrete concepts 4 images included more than one individual. However, in 3 extra cases of concrete concepts the images implicitly referred to other people: 1) in one, an old lady visited dead people at the cemetery, 2) in another, a girl was communicating through the telegraph, 3) in the third, a woman seemed to invite someone to sit and eat. Second, we considered the number of people in the images. The number of people presented in the concrete images greatly outnumbers that presented in the abstract ones. Indeed, in the case of concrete concepts there are four cases (army, float, camping and canoe) in which many people are present.

Moreover, in order to fully address the Reviewer comment, we have now conducted an on-line survey including 29 participants (age= 43.38 st dev= 14.24; 18 females), which were asked to rate the social component present in each picture by using a Likert scale ranging from 0 to 7 points. The frequency distribution of the rating about the presence of the social component attributed results to be balanced between abstract and concrete concepts, as testified by the non-significant Chi Square ($X^2(df=6)=5.51, p=.88$). We have now added the excel file with all the data in the OSF link <https://osf.io/98q2g>.

Based on these considerations, we believe that our effects are not due to the fact that the content of abstract concepts refers more to sociality than that of concrete ones.

We have now added the results of the survey also in the manuscript lines (176-181) pag (6) :

- ❖ *“Moreover, to control that the abstract and the concrete pictures were equally representing social contexts, we have conducted another on-line survey including 29 participants, which were asked to rate the social component present in each picture by using a Likert scale ranging from 0 to 7 points. The frequency distribution of the rating about the presence of the social component attributed results to be balanced between abstract and concrete concepts, as testified by the non-significant Chi Square ($X^2(df=6)=5.51, p=.88$).”*

3.) The mechanisms underlying the putative carry-over effects of the guessing task to the avatar interaction task should be better explained in the introduction section. This is particular important because the two tasks are not directly related in their conceptual content, besides the presence of the confederates. What is the theoretical justification to expect such carry-over effects? Which role does the helpfulness of the confederate play?

- ❖ We thank the Reviewer for asking to make more explicit the rationale of the study, which is indeed built on separate methodological sessions, and to strengthen the theoretical assumption on the carry-over effect. As the Reviewer correctly claimed, the presence of the confederates is the element of continuity between the Concept Guessing Task and the Joint Grasping Task.

The confederates in the experimental context represented two referents who were cooperating at different levels (“cognitively” during the Concept Guessing task and in a sensorimotor way during the Joint Grasping one) with the participant in order to successfully reach two sort of objectives (intellectual and sensorimotor). The novelty of the WAT theory is to point out the importance of the social interaction in the acquisition of abstract more than concrete concepts. The hypothesis of the present work is that experiencing the necessity to take advantage from another human in order to reach an intellectual objective might favor the subsequent attitude to establish a satisfactory sensorimotor interaction with him/her. Participants who have benefited from the other’s help to guess an abstract meaning might develop the implicit willingness to create a successful relationship with him/her, knowing that the other might help them in the following session. Importantly, the confederates did not take any active role in the sensorimotor task that may have biased the participant’s behavior, since they were replaced by a fixed avatar during the Joint Grasping task. We believe that interacting with an avatar during the Joint Grasping task matched a real interaction, as indicated by a large literature (e.g. Garau et al., 2005; Sacheli et al., 2015; Osimo et al., 2015; Moreau et al., 2020). A successful sensorimotor interaction between the avatar and the participant corresponded with a successful sensorimotor interaction between the confederate and the participant.

- ❖ We have now added this passage in the introduction lines (101-105), pag (4) :

“Our idea is that being helped by another human in order to reach an intellectual objective might favor the subsequent attitude to establish a satisfactory sensorimotor interaction with him/her. Participants who have benefited from the other’s help to guess an abstract concept might develop the implicit willingness to create a

successful relationship with him/her, knowing that the other might help them in the following session.”

I also missed an analysis elucidating the role of the helpfulness of the confederate, which could affect synchrony in the avatar interaction task.

- ❖ **The analysis on the direct impact of the other’s help on participants’ performance at the motor interaction task is reported in the supplementary materials. It includes both the Subjective and Objective need of other’s help indexes as continuous predictors in the linear mixed models on Grasping Asynchrony. Here is the description of the results of these analyses:**

“Grasping Asynchrony

- ❖ *The R²c of the model is = 0.23. This model yielded a significant main effect of the Type of Concept (Abstract vs Concrete), ($F(1,3689.148)=8.59, p=.001$) due to the fact that abstract Type of Concept showed less Grasping Asynchrony (mean = 148, SE= 9.04) compared with concrete Type of Concept (mean = 158, SE=9.03). A significant three way interaction between the Type of Concept (Abstract vs Concrete), the Interaction Type (Imitative vs Complementary) and the Objective need of other’s help ($F(1,3689.229)=7.05, p=.01$) was also found. Simple slope analysis showed that the slopes of Concrete Type of Concept in the Imitation condition was significantly different from zero as a function of the Objective Need of other’s Help [LCI 11.9- UCI 66.2]. The pairwise difference between the simple slopes of movements associated to Abstract and Concrete concepts during the Imitative condition as a function of the Objective Need of other’s Help was significant (estimate = -22.15, SE = 6.87, $t(3689)=3.223, p=.01$). In the Imitative condition and in the Abstract concept block, participants showed more synchrony with the avatar’s action in comparison with the Concrete concept block and their better performance was modulated by the Objective Need of other’s Help index. The more participants were asking suggestions to the confederate (higher Objective Need of other’s Help), the more they were synchronized with the confederate avatar’s action. A significant three way interaction between the Type of Concept (Abstract vs Concrete), the Interaction Type (Imitative vs Complementary) and the Subjective need of other’s help index ($F(1,3689.087)=5.54, p=.02$) was also reported. Simple slope analysis showed that none of the slopes was significantly different from zero. The pairwise difference between the simple slopes of movements associated to Abstract and Concrete concepts during the Imitative condition as a function of the Subjective Need of other’s Help index was significant (estimate = .9004, SE = 0.207, $t(3689)=4.34, p<.001$). Finally, there was a significant three way interaction between the Type of Concept (Abstract vs Concrete), the Interaction Type (Imitative vs Complementary) and the Movement Type (Precision vs Power Grip) ($F(1,3689.030)=4.89, p=.027$). The Relevant Post-hoc tests were not significant.”*

We have now added a discussion section in the supplementary materials lines (909-934) pages (36):

- ❖ *“In the Imitative condition and in the Abstract concept block, participants showed more synchrony with the avatar’s action in comparison with the Concrete concept block and their better performance was modulated by the Objective Need of other’s Help index. The more participants were asking suggestions to the confederate (higher Objective Need of other’s Help), the more they were synchronized with the confederate avatar’s action. On the contrary, the smaller the Subjective Need of other’s Help index for Abstract than Concrete Type of Concept, the larger the difference in the grasping asynchrony between the Abstract and Concrete blocks. In other words, the less participants reported the Subjective Need of the other’s help, the more they were synchronized with the avatar’s action in the Abstract compared to the Concrete block. Although counterintuitive, this result shows that the Subjective and Objective need of other’s help differently influences participants’ performance at the interactive task. In line with this finding, there was an absence of correlation between the Subjective need of other’s help index and the Objective need of other’s help index ($\rho=.09$, $p=.71$). Specifically, the higher was the Objective need of other’s help, the more participants were synchronous with the avatar associated with the confederate in charge to provide hints to guess Abstract vs Concrete concepts. On the other side, the higher was the Subjective need of other’s help, the less participants were synchronous with the avatar associated with the confederate in charge to provide hints to guess Abstract vs Concrete concepts. The above-mentioned dissociation indicates that an objective index, such as the Objective need of other’s help, exerts an opposite modulation on the interactive task compared with a subjective index, such as the Subjective need of other’s help. One possibility is that the self-perception about the owned conceptual knowledge required during the guessing task, could have been misinterpreted by participants. They, indeed, might have not experienced the gap between their own knowledge and the required knowledge consciously, or still explicitly have denied such a gap. In this way, only the Objective index may have captured the actual need of the other’s help.”*

I also had problem to link hypothesis c) about automatic imitation with the main aim of the study.

- ❖ **We acknowledge that this hypothesis may have not been clearly explained in the previous version of the manuscript. We try to do it here and in the new version of the manuscript. As far as the hypothesis on automatic imitation is concerned, it has been shown that the spontaneous tendency to imitate other’s motor behavior influences the quality of social interaction we engage in (Salazar Kamp et al., 2017). More specifically, we tend to engage in more positive social interaction with people we imitate or by whom we are imitated (Salazar Kamp et al., 2017). Here we hypothesize that in the attempt to promote a positive social interaction during the Concept Guessing Task and in particular when required to guess abstract concepts, participants would imitate the other’s action more. We now extended on this aspect in the introduction:**

- ❖ c) *“We expect participants to show more automatic imitation (visuo-motor interference effects) of the ACs avatar. Indeed, it has been shown that the automatic and unconscious imitation of other’s movements creates a positive social relationship between interacting agents (Salazar Kamp et al., 2017). The human-avatar motor interaction task used in this study has been shown to elicit automatic imitation when participants perform complementary interactions both with a virtual and a human partner (Candidi et al., 2017; Gandolfo et al., 2019; Moreau et al., 2020; Sacheli et al., 2012; 2015). Using the same task, it has also been shown how automatic imitation is influenced by the social relationship between interacting people: participants imitate less their partner’s movements when they have a negative relationship with him/her or when interacting with an out-group partner, if they have a negative implicit bias towards the out-group (Sacheli et al., 2012; 2015). Here we hypothesize that, in the attempt to promote a positive social interaction, participants would imitate the other’s action more. Establishing a positive relationship could, namely, be fruitful during the Concept Guessing Task, in particular when participants will have to guess abstract concepts.”*

Minor points

4.) Were t-tests two-tailed or one-tailed?

- ❖ **T-test were two tailed, we have now specified this info in the Result session.**

5.) Were confederates physically present during the guessing tasks or just simulated on the computer?

- ❖ **Only one confederate, the one in charge of delivering hints about abstract or concrete concepts was physically present in the room during the guessing tasks, the other one was waiting her turn to take part in the following experimental session.**

We have now added this information in the manuscript lines (230-232) pag (8):

- ❖ ***“Only one confederate, the one in charge of delivering hints about abstract or concrete concepts was physically present in the room during the guessing tasks, the other one was waiting her turn to take part in the following experimental session.”***

6.) The authors missed to quote in the introduction earlier work proposing a multiple representation theory of abstract concepts, which emphasizes the role of social interactions, in addition to sensory-motor, emotional, introspective and linguistic information (Kiefer, M., & Harpaintner, M. (2020). Varieties of abstract concepts and their grounding in perception or action. *Open Psychology*, 2, 119-137.). The authors should also acknowledge the heterogeneity of the meaning of abstract concepts as suggested by earlier work (Ghio, M., Vaghi, M. M. S., & Tettamanti, M. (2013). Fine-grained semantic categorization across the abstract and concrete domains. *Plos One*, 8(6); Harpaintner, M., Trumpp, N. M., & Kiefer, M. (2018). The semantic content of abstract concepts: A property listing study of 296 abstract words. *Frontiers in Psychology*, 9, 1748; Hoffman, P. (2016). The meaning of 'life' and other abstract words: Insights from neuropsychology. *Journal of Neuropsychology*, 10(2), 317-343). Finally, the authors should also mention that the

importance of visual and motor information for abstract concepts has been demonstrated in a recent fMRI study (Harpaintner, M., Sim, E. J., Trumpp, N. M., Ulrich, M., & Kiefer, M. (2020). The grounding of abstract concepts in the motor and visual system: An fMRI study. *Cortex*, 124, 1-22.).

- ❖ **We thank the Reviewer for mentioning this literature that we now discuss in the paper. As suggested, we mentioned the paper by Harpaintner et al., 2020 that recently highlighted the role of visual and motor information for abstract concepts. Furthermore, we briefly mentioned that recently authors have started to investigate differences among types of abstract concepts, and mentioned all the literature suggested by the Reviewer. Lines (61-77) pag (3)**

“According to embodied theories both CCs and ACs were grounded in sensorimotor experience. Theories of distributed semantics, which intended meaning as given by associated words, ascribed a major relevance to language. Recently hybrid views, such as the multiple representation ones, emerged (review: Borghi et al., 2017). **According to them, ACs would be grounded in sensorimotor systems (for recent evidence on the importance of visual and motor information, see Harpaintner et al., 2020)**, like CCs, but they would activate to a larger extent linguistic, emotional and social experience. While there is plenty of evidence on the role played by both linguistic (Borghi et al., 2019; Dove, 2014; 2019; Recchia & Jones, 2012) and emotional experience (Vigliocco et al., 2014) for ACs, social experience has not received the same attention. **Only very recently, some authors have started to investigate fine-grained differences among types of abstract concepts, without considering them as an homogeneous group (e.g. Ghio et al., 2013; Harpaintner et al., 2018; Hoffman, 2016; Muraki et al., 2020; Villani et al., 2019). Importantly for us, some recent studies have focused on the neural representation of abstract social concepts comparing them with other concepts (Desai et al., 2018; Mellem et al., 2016). Within multiple representation theories on ACs, the Words As social Tools (WAT) proposal has put a special emphasis on sociality (Borghi & Binkofski, 2014; Borghi et al., 2019). This emphasis is consistent with data showing that ACs, compared to CCs, evoke more introspective and social features (Barsalou & Wiemer-Hastings, 2005; Harpaintner et al., 2018; Kiefer & Harpaintner, 2020).”**

7.) line 361: “In line with our hypothesis (a), participants asked more suggestions and committed more errors” This statement is not correct because the Objective Helping index, to which this statement refers, is calculated as RATIO between number of requested suggestions and accuracy.

- ❖ **We agree with the Reviewer that the statement should be better re-formulated in order to cast out any doubts about the consistency between the theoretical expectations and the dependent variables we computed. Anyway, the Objective Helping index numerically expresses the difficulty encountered by the participants during the concept guessing task, such difficulty is captured by the averaged suggestions requested and by the averaged accuracy of responses. The two values are related, and it makes sense to consider them in a unique index, indeed the number of suggestions per se is not informative about the performance of the participant, since she could**

either have asked for many suggestions without guessing the concept at the end, or have asked for many suggestions to finally guess the concept. For the sake of clarity, we reported the number of suggestions requested (abstract concepts= 632; concrete concepts= 444, $t(20)=-6.09$ $p<.001$), the averaged suggestions requested (abstract concepts= 2.08 st dev= .62; concrete concepts=1.11 st dev=.23 $t(20)= 10.15$ $p<.001$) and the accuracy (abstract concepts= 0.75, st dev= .08; concrete concepts= 0.94, st dev=.07 $t(20)=-11.74$ $p<.001$). The two-tailed t-test comparisons show that all the three measures significantly differ between abstract and concrete concepts.

- ❖ **We have now changed the sentence in the manuscript lines (393-395) (pag 13) and in the Figure 4 caption: "*Participants showed a higher value of the Objective Helping index, which is the ratio between the averaged suggestions required and the averaged accuracy.*"**

8.) The authors write on p. 18 that the study is preregistered on <https://osf.io/98q2g/>. Unfortunately, I did not find the preregistered study protocol describing the goal of the study, hypotheses, methods and data analyses, but only some data files and one R script including LMM analyses. When clicking in osf on the registrations button, I only receive the message "There have been no completed registrations of this project. Perhaps I missed to find the location of this information on the osf website. I also did not find the data files containing the Objective and subjective Helping indices and the R-scripts for the t-tests and analyses described in the supplement.

- ❖ **We are very sorry for this inconvenience. At the following link <https://osf.io/4tbme>, the Reviewer hopefully can find the preregistered report, and at <https://osf.io/98q2g> there are two folders, one folder named "Materials" with the stimuli inside, and a folder named "Data" with the R scripts used for each variable analyzed. We have performed t-tests with the software "Statistica", thus we do not have the R scripts concerning such analyses. However, we have added now the excel file with the data on the Subjective (VAS) and Objective Helping indices (indices_file).**

9.) line 474: The authors discuss visuo-motor interference effects influenced by subjective need of help. However, the analyses were only reported in the quite lengthy supplement (which might be shortened). If this finding is deemed important, the corresponding analyses should be moved to the main text.

- ❖ **Thank you for highlighting this point. We agree that the finding of the influence of other's help on visuo-motor interference was not in our main hypotheses and that the lengthy supplementary materials may not allow readers to fully understand our analyses. For this reason, we removed the comment on the influence of other's help on visuo-motor interference from the discussion and we removed the related analyses from supplementary materials.**

Reviewer: 2

Comments to the Author(s)

In this paper, the authors investigate the relationship between concept abstract/concreteness and sociality, by measuring the degree to which participants coordinate their actions with confederates. One confederate gave hints to a concrete concept (CC); the other to an abstract (AC). Coordination was measured in a motor interaction task with avatars corresponding to the confederates. There was more coordination with the AC hinting cofederate, suggesting a link between conceptual abstraction and sociality or recruitment of others.

I think that this is an ingenious idea and a well executed experiment and analysis, and I commend the authors. My main comment, which I think needs to be addressed before publication, concerns a confound which I think questions the claims that the wish to make.

My worry is that the difference in behaviour towards AC and CC confederates is being driven by task difficulty alone.

We are grateful to the Reviewer for the positive comments and to give us the opportunity to address this issue.

Imagine a task with maths multiple choice questions (or general knowledge questions). There were easy and hard categories (that we can objectively assess, in the way that AC and CC questions had an objective difference in difficulty in this experiment). The two confederates only give hints to the easy or the hard maths/general knowledge questions respectively.

Isn't it very plausible, in this task, that the participant will become more behaviourally coupled to the confederate giving the hints to the hard question. This is because that confederate is seen as smarter, more valuable in this context, more pro social and helpful, etc.

If we did get that result in my imaginary task, it would suggest that such differences in motor coordination are telling us about how people recruit others in the service of more difficult tasks, by recruiting behavioural coupling. That's a really neat finding! But it would mean that the results the authors present here don't tell us anything about concepts and their structure, or the social nature of ACs vs CCs.

So, to accept the authors current conclusions, I would want to see data from a task like this, mashed as closely as possible to the conceptual one, where there were similarly two levels of difficulty, but with no relation to concept concreteness. If behavioural coordination was modulated exclusively in the conceptual task, then I would buy the conclusion that it is the social content of ASc specifically that is the reason they modulate motor coupling.

- ❖ **We thank the Reviewer for raising this point, that allows us to address more directly the “difficult” issue of the difficulty confound. Notice, however, that we are not claiming that our results pertain to abstract concepts for their social content. We actually believe that it is exactly because of the complexity of ACs that we rely more on other people. We actually agree that the effect we found is associated with abstract concepts being more difficult to guess than Concrete Concepts and that, by virtue of this, participants need to rely more on interpersonal help in our task. We also believe that the point raised by the Reviewer captures a core problem of abstract (vs) concrete concepts: they might be intrinsically more difficult to learn, to guess and to use than concrete ones. At the same time, matching CCs and ACs concepts for difficulty would be problematic, since we would risk to select bad examples of both. Somehow, we believe that there is no control experiment we can perform. However, the point raised by the Reviewer is a challenge and we definitively believe that this is an interesting issue for future experiments. Following the suggestion of the Reviewer, we have mitigated our conclusions, because we can “only” claim that we found that abstract concepts elicit more pro-social behaviors than concrete ones, but we cannot exclude that this result does not extend to other domains.**

We have now added this passage in the discussion: (lines 534-564) pages (17-18):

“In this work we have demonstrated that with ACs participants rely more on other people during their guessing than with CCs. Which mechanism subtends this phenomenon? One could object that the effect we found could be due to the fact that abstract concepts are more difficult than concrete ones to guess/learn. Because they are more difficult, we would need to rely more on other people, hence we would be more collaborative with them.

This is actually what we think. Difficulty is an intrinsic feature of abstract concepts, when compared to concrete concepts. First of all, abstract concepts are more difficult to form, because their examples are more different from each other than those of concrete concepts. Furthermore, abstract concepts are more difficult to process and recall, as revealed by the well-known concreteness effect, i.e., the advantage of concrete words in processing and recognition (e.g. Paivio et al., 1994). Finally, abstract concepts are also explicitly perceived by participants as more difficult overall. In a recent rating study we asked participants to simply evaluate the “difficulty” of the written words on a 7-point scale; participants were assigned to different interfering conditions (Villani et al., 2020). The conditions influenced the ratings, but across the conditions abstract concepts were always considered more difficult than concrete ones.

The notion of difficulty accounts for the particularity of ACs: ACs are difficult because they are more detached from sensorimotor experience than CCs, because they assemble multiple and diverse members without perceptual similarity, because they do not have a single clearly delimited referent, and finally because they are acquired later than concrete ones and mostly through language.

In sum: we think to have demonstrated that, because of their difficulty, ACs elicit more pro-social behaviors than CCs. This important objection, the fact that the effect mainly depends on difficulty, might however lead to very fruitful research. Does the effect extend beyond the guessing task? We have good reasons to believe that it does and that it involves more generally the use of abstract concepts, for the aforementioned reasons. Does the effect we found with abstract concepts extend also to other difficult concepts and situations? Do we tend to be more collaborative with others, when faced with complex problems that the others can help us solve? Further research is needed, to address these questions.”

References

- Borghì, A. M., Barca, L., Binkofski, F., & Tummolini, L. (2018). Abstract concepts, language and sociality: from acquisition to inner speech. *Philosophical Transactions of the Royal Society B: Biological Sciences*, 373(1752), 20170134.
- Sacheli, L. M., Christensen, A., Giese, M. A., Taubert, N., Pavone, E. F., Aglioti, S. M., & Candidi, M. (2015). Prejudiced interactions: implicit racial bias reduces predictive simulation during joint action with an out-group avatar. *Scientific reports*, 5, 8507.
- Osimo, S. A., Pizarro, R., Spanlang, B., & Slater, M. (2015). Conversations between self and self as Sigmund Freud—A virtual body ownership paradigm for self counselling. *Scientific reports*, 5, 13899.**
- Garau, M., Slater, M., Pertaub, D. P., & Razzaque, S. (2005). The responses of people to virtual humans in an immersive virtual environment. *Presence: Teleoperators & Virtual Environments*, 14(1), 104-116.**
- Moreau, Q., Candidi, M., Era, V., Tieri, G., & Aglioti, S. M. (2020). Midline frontal and occipito-temporal activity during error monitoring in dyadic motor interactions. *Cortex*, 127, 131-149
- Salazar Kämpf, M. et al. (2017). Disentangling the Sources of Mimicry: Social Relations Analyses of the Link Between Mimicry and Liking. *Psychological Science*, 29, 131–138.
- Ghio, M., Vaghi, M. M. S., & Tettamanti, M. (2013). Fine-grained semantic categorization across the abstract and concrete domains. *Plos One*, 8(6)**
- Harpaintner, M., Trumpp, N. M., & Kiefer, M. (2018). The semantic content of abstract concepts: A property listing study of 296 abstract words. *Frontiers in psychology*, 9, 1748.**
- Harpaintner, M., Sim, E. J., Trumpp, N. M., Ulrich, M., & Kiefer, M. (2020). The grounding of abstract concepts in the motor and visual system: An fMRI study. *Cortex*, 124, 1-22.**
- Hoffman, P. (2016). The meaning of ‘life’ and other abstract words: Insights from neuropsychology. *Journal of neuropsychology*, 10(2), 317-343.**
- Kiefer, M., & Harpaintner, M. (2020). Varieties of abstract concepts and their grounding in perception or action. *Open Psychology*, 2(1), 119-137.**
- Paivio, A., Walsh, M., & Bons, T. (1994). Concreteness effects on memory: When and why?. *Journal of Experimental Psychology: Learning, Memory, and Cognition*, 20(5), 1196.**
- Muraki, E. J., Sidhu, D. M., & Pexman, P. M. (2020). Heterogenous abstract concepts: is “ponder” different from “dissolve”? *Psychological Research*, 1-17.**
- Villani, Caterina, Luisa Lugli, Marco Tullio Liuzza, and Anna M. Borghi. (2019) "Varieties of abstract concepts and their multiple dimensions." *Language and Cognition* 11, no. 3 (2019): 403-430.**

Appendix B

Review

I have read the paper myself and we have received further comments from one reviewer. I agree with both reviewers' initial evaluations that your work on the grounding of abstract concepts in sociality is interesting and novel, and it adds to the literature.

Previously, both reviewers had concerns about the work, especially about the level of difficulty. Reviewer 1 also wanted to see the rating and performance data; both they and I feel that this has not been properly addressed in your reply to the comments or in the manuscript. I would like you to further address the issue of difficulty and rating/performance data, as per Reviewer 1's additional comments. I also have a few minor points in the methods which need addressing:

We thank Editor Ackerley for the positive comments about the manuscript. We hope that the additional surveys and experiments that we have now run will be considered satisfactory to address the crucial points raised by the Editor and Reviewer 1. We did our best to address the concern on the relation between the images employed in the experiment and the abstract/concrete nature of the concepts. Regarding the difficulty issue, we can only provide additional theoretical arguments to sustain that the “difficulty” is an intrinsic property of abstract concepts that cannot be “surgically” extracted from abstract concepts without altering their nature.

- Did the participants give written informed consent?

- ✓ **We thank the Editor for asking for this information. Each participant signed the informed consent before starting the experiment, as we have now specified on *pages 5-6 (Lines 143-145): “The study was in accordance with the Declaration of Helsinki and approved by the ethical committee of Sapienza University of Rome, Department of Dynamic and Clinical Psychology, and Health Studies. Before starting the experiment, each participant was asked to sign the informed consent approved by the ethical committee of Sapienza University of Rome, Department of Dynamic and Clinical Psychology, and Health Studies.”***

- The authors have now added ‘The analyses including the covariates have been moved to the supplementary materials instead of being part of the main text’. This phrasing is a little strange here and it would be far simpler to say something like: The analyses including the covariates can be found in the supplementary materials.

- ✓ **We thank the Editor for this suggestion. We have now substituted the previous sentence with the simpler one indicated by the Editor, on page 5 (Lines 136-137): “The analyses including the covariates can be found in the supplementary materials.”**

- The text that has been added on p.6 of the methods seems to be in the wrong tense, for example, ‘we have conducted’. This can all be corrected by removing ‘have’ for each occasion in this new text.

- ✓ **We thank the Editor for highlighting such imprecision; we have now corrected the entire paragraph by removing “have” where it occurred. Pages 6-7 (Lines 170-184).**

Reviewer comments to Author:

Reviewer: 1

Comments to the Author(s)

This is a revised manuscript, which I have already evaluated previously. Overall, the authors have improved the manuscript in several respects, as suggested by the reviewers. However, the authors were not responsive with regard to some relevant aspects. Several concerns therefore remain and require further improvements

- ✓ **We are glad that Reviewer 1 found the manuscript improved. We hope that Reviewer 1 will agree that we tried to dispel any doubts with additional empirical tests to support a theoretical argument about abstract concepts' nature.**

1) I still would like to see rating and performance data, i.e. performance in a semantic relatedness judgment task, which indicate that semantic relatedness/task difficulty of the selected pictures were matched for ACs and CCs. The new control study using a guessing task is not sensitive enough to reveal differences in the stimulus pairs between ACs and CCs.

- ✓ **We thank Reviewer 1 for raising the issue about the validity of the stimuli. We agree that it is indeed a crucial aspect to address, both empirically and theoretically. As suggested by Reviewer 1, we performed a semantic relatedness judgment task with the stimuli used in the experiment and with other new 40 stimuli (in two different experiments) (20 abstract, 20 concrete) selected from the same database (Della Rosa et al., 2010). You can find the new images at the OSF link <https://osf.io/98q2g>.**

Before presenting the additional experiments, we would like to specify some points that, in our view, deserve to be theoretically clarified.

The guessing task was structured to be not time-sensitive but to measure the effort required by each participant to guess the concepts based on the asked hints. To guess abstract concepts, participants required more hints than to guess concrete ones, as the objective helping index testifies. We agree with the Reviewer that it is crucial to understand whether the results could be driven by the selected stimuli per se or due to abstract concepts' intrinsic characteristics. Following this reasoning, we performed an online survey (reported in the previously revised paper), asking participants to guess the concepts our images were referring to (similarly to what happened in the actual experiment). We demonstrated that participants achieved a similar performance when required to guess abstract and concrete concepts. Even if the performance with concrete and abstract concepts did not differ in this online guessing task, participants might experience more difficulties associating images to abstract concepts. We can discover this employing a task that considers timing, as the semantic relatedness task suggested by the reviewer.

Unfortunately, validating a dataset of images representing abstract concepts requires overcoming some aspects that are rooted in their nature. Compared to concrete concepts, abstract concepts are, namely, less imageable (Paivio, 1990), more detached from sensorial modalities (Barsalou, 2003), even if still grounded in sensorimotor and affective properties (Harpaintner et al., 2020a). Furthermore, they are acquired later, mostly through the linguistic modality (Wauters et al., 2003; Della Rosa et al., 2010; Villani et al., 2019) (Borghi & Binkofski, 2014; Borghi et al. 2019). Consequently, they are more flexible (Harpaintner et al., 2020b), and they refer to a

multitude of contexts (Contextual Availability: Schwanenflugel et al., 1992), which means that many situations and sensations can represent them. Furthermore, abstract concepts are more heterogeneous since their members have less common features (low-dimensionality: Lupyan & Mirman, 2013; Borghi, 2020) and are therefore more variable across and within individuals and cultures.

To make an example, selecting an image to represent the concept of “beauty” is not the same as selecting an image of the concept of “glass”. “Beauty can be different things, it is in the eye of the beholder” (see Harpaintner et al., 2020a); therefore, it might be related to different sensorial, social, cultural, and linguistic experiences.

We can even arrive at the point in the abstractness continuum where an abstract visual representation evokes in each beholder a different concept. The challenge of abstract concepts consists of understanding how and to what extent they are grounded even if they miss an object as a referent, are very heterogeneous, and are less imaginable.

Crucially, abstractness and imageability are highly correlated, and for many years they have been treated as equivalent constructs (Kousta et al., 2011). Even if not equivalent, the more the concepts grow in abstractness, the less they are imaginable. Hence, we might fail to preserve the object of our research by selecting concepts that can be very easily represented by images and thus poorer in abstractness.

However, we agree with Reviewer 1 that an additional effort to prove the validity of the stimuli is useful to improve the quality of the paper. Intending to shed light on the issue raised by Reviewer 1, we conducted a first survey, in which we asked participants to rate through a Likert scale ranging from 0 to 7 how much the presented images were representative of the associated concept. In the first survey, 28 participants (Mean age= 35.67 st dev=12.20, 21 F) evaluated the 40 images employed in the experiment (20 abstract, 20 concrete). The frequency distribution of the rating (from 0 to 7) was not balanced between abstract and concrete concepts, as testified by the Chi Squared analyses ($X^2(df=7)=80.91, p=.000$). Specifically, participants judged as more representative the images of concrete than abstract concepts. You can find the data file at the OSF link: <https://osf.io/98q2g>. In order to verify whether this unbalance was due to the specifically selected stimuli or instead to the lower imageability of abstract compared to concrete concepts, we run a second survey. We selected other 40 (20 abstract, 20 concrete) concepts from the database of Della Rosa et al. (2010), balanced in familiarity (Abstract concepts mean= 499.01, dev st= 71.95; Concrete concepts mean= 532.88, dev st= 98.43) ($t(38)=-1.24, p=.22$), and word length (Abstract concepts mean =7.75, dev st= 2.17; Concrete concepts mean= 6.9, dev st=2.10) ($t(38)=1.18, p=.24$).

We then adopted the same procedure used in the main experiment. First of all, we asked a new sample of 14 participants to write 6 contexts/situations related to each concept. Five participants did not complete the task; thus, the final sample was composed of 9 participants (Mean age= 40.66 st dev=16.49, 7 F). From the most frequent situations produced, three experimenters selected the new set of 40 images. Then, we asked other 29 participants (Mean age= 40.89 st dev=11.84, 21 F) to rate through a Likert scale ranging from 0 to 7 how much each image was representative of the associated concept. The frequency distribution of the rating (from 0 to 7) was not balanced between abstract and concrete concepts, as testified by the Chi Squared analyses ($X^2(df=7)=55.56, p=.000$). Again, participants judged as more representative the images representing concrete rather than abstract concepts. You can find the data file in the OSF link: <https://osf.io/98q2g>.

After performing the two surveys, which converged on the same results, we performed the semantic relatedness judgment task suggested by Reviewer 1. We used both the “old experimental” images and the new ones. To smoothen the difference between abstract and concrete concepts, the object was displayed inside a social context, similarly to what we had done in the main experiment, in all but three images on concrete and abstract concepts.

In a first experiment, we included the “old experimental” 40 images (20 abstract, 20 concrete) and other 12 new images (six abstract, six concrete) from Della Rosa et al., 2010. We entered these new ones to have a preliminary idea of whether the “old experimental” 40 images were less valid than these new few ones.

We run the experiment by using the online platform Gorilla (www.gorilla.sc; Anwyl-Irvine et al., 2019). We presented images combined with the correct corresponding word (congruent trials) or not. In total, there were 26 congruent concrete trials, 26 congruent abstract trials, and 20 incongruent trials, which were not considered in the analysis. Each trial started with the presentation of a fixation cross lasting 400 ms at the center of the screen, followed by a stimulus lasting 2500 msec with the image located in the middle of the screen and the associated word below it. Twenty participants (Mean age= 31.15, st dev= 6.09, 12 F) were asked to clicking with the mouse on two different buttons depending on whether the image and the word were associated or not.

A visual countdown starting from 1000 msec before the end of the stimulus was entered to prompt the response. After having removed the errors (i.e. abstract 104 (17.99 %), concrete 41 (7.93 %), and having considered only RTs included in ± 2 st dev from the general average, the analysis was restricted to 841 RTs out of trials 1035 (5 trials were not registered). Such percentage discrepancy in the number of errors ($X^2(df=1)=31.69$, $p=.000$) clearly indicates that the abstract concepts were more difficult to associate with the corresponding images. In the mixed model on RTs, we included as fixed effects the Category of the concept (abstract, concrete), the Type of images (old, new), and their interaction, and we added random intercepts for Subjects and the Stimuli. The model yielded a significant main effect of the Category ($F(1,46.799)=9.6389$, $p<.003$): overall abstract concepts showed slower RTs (1477, $SE= 48$) than concrete ones (1339, $SE=47.6$). Crucially, there was not a main effect of the Type of images ($F(1,46.667)=0.2953$, $p=0.58946$), nor an interaction between the two fixed factors ($F(1,46.651)=0.0250$, $p=0.87517$). These results suggest that the abstract images are less representative of the abstract concepts. Importantly, this result does not seem to be due to the specific images selected for the experiment.

To reach a stronger conclusion, we tested other 20 participants (Mean age=31.15;st dev= 6.39, 17 F) who performed the same semantic relatedness judgment task with the entire new dataset of 20 abstract and 20 concrete stimuli. In total, there were 20 congruent concrete trials and 20 congruent abstract trials; the 12 incongruent trials were not considered in the analysis. After removing the errors (i.e. abstract 47 (11.75 %), concrete 27 (6.75 %)) and considering only RTs included in ± 2 st dev from the general average, the analysis was restricted to 694 RTs out of trials 800. Again, the percentage discrepancy in the number of errors ($X^2(df=1)=5.96$, $p=0.014$) indicates that the abstract concepts were more difficult to associate with the corresponding images. In the mixed model on RTs, we included as fixed effects the Category of the concept (abstract, concrete), and we added random intercepts for Subjects and the Stimuli. The model yielded a significant main effect of the Category ($F(1,37.93)=5.59$, $p<.03$) since overall abstract concepts showed slower RTs (1463, $SE= 42$) than

concrete ones (1404, SE= 41.7). The abstract images are thus less representative of the abstract concepts also in this new dataset.

Moreover, we computed a difference between RTs in the abstract and concrete concepts condition (Δ RTs) separately for the old and new stimuli. We then compared RTs of the old and new stimuli using an independent sample t-test. Results showed that RTs did not differ between the old and new stimuli ($t(38) = -0.87, p = 0.47$). Thus the difference between abstract and concrete concepts is present both in the old and new stimuli.

In conclusion, we thank Reviewer 1 for suggesting these further controls. We think that the consistent pattern of results obtained in the two experiments strongly suggests that the lower association with images is an intrinsic property of abstract compared with concrete concepts. Eliminating it would be an experimental artifact that does not consider these two kinds of concepts' particularities. Notably, a recent study (Lakhzoum et al., 2020) arrives at similar conclusions, showing that facilitation of related over unrelated picture-word combinations is stronger with concrete than with abstract stimuli.

We now added these surveys in Supplementary Materials: Lines 845-946 pag (33-35)

2.) The authors now argue in the discussion section (p. 17) that ACs are more difficult, so that individuals would rely more on other people and would be more collaborative with them. If this is a mere effect of difficulty, the same effect should also show up for more difficult CCs. To support their claim, the authors could run an analysis on CCs with conceptual difficulty as additional factor.

- ✓ We thank the Reviewer for the suggestion. Unfortunately, we cannot run the suggested analysis because the joint action task was performed after sessions of conceptual guessing task composed of entire blocks of abstract or concrete concepts, so the single concept's effect cannot be taken into consideration. We reasoned that for participants to feel they need the other more when guessing abstract concepts, they needed to perform the guessing task with different concepts.

We agree with the reviewer that it is important to exclude that our results were due to mere difficulty. See the passage added in the manuscript in the first step revision process: (lines 534-564) pages (17-18).

Importantly, however, we hope to have shown that our results are due to the difficulty of stimuli per se, but rather to a kind of difficulty associated intrinsically with abstract concepts. Theoretically, such “difficulty” derives from the combination of many parameters that characterize abstract concepts such as the scarce imageability, the late age of acquisition, the linguistic modality of acquisition, the stronger need for other’s help (Villani et al., 2019). Eliminating such intrinsic difficulty would be working with materials that do not reflect human natural categories.

References

- Anwyl-Irvine, A., Massonnié, J., Flitton, A., Kirkham, N., & Evershed, J. (2018). Gorillas in our Midst: Gorilla. sc, a new web-based Experiment Builder. *Behavior Research Methods*, <https://doi.org/10.3758/s13428-019-01237-x>
- Barsalou, L. W. (2003). Abstraction in perceptual symbol systems. *Philosophical Transactions of the Royal Society of London. Series B: Biological Sciences*, 358(1435), 1177-1187.
- Borghì, A. M. (2020). A future of words: Language and the challenge of abstract concepts. *Journal of Cognition*, 3(1).
- Borghì, A. M., Barca, L., Binkofski, F., Castelfranchi, C., Pezzulo, G., & Tummolini, L. (2019). Words as social tools: Language, sociality and inner grounding in abstract concepts. *Physics of life reviews*, 29, 120-153.
- Borghì, A. M., & Binkofski, F. (2014). *Words as social tools: An embodied view on abstract concepts* (Vol. 2). Springer New York.
- Della Rosa, P. A., Catricalà, E., Vigliocco, G., & Cappa, S. F. (2010). Beyond the abstract—concrete dichotomy: Mode of acquisition, concreteness, imageability, familiarity, age of acquisition, context availability, and abstractness norms for a set of 417 Italian words. *Behavior research methods*, 42(4), 1042-1048.
- Harpaintner, M., Sim, E. J., Trumpp, N. M., Ulrich, M., & Kiefer, M. (2020a). The grounding of abstract concepts in the motor and visual system: An fMRI study. *Cortex*, 124, 1-22.
- Harpaintner, M., Trumpp, N. M., & Kiefer, M. (2020b). Time course of brain activity during the processing of motor-and vision-related abstract concepts: flexibility and task dependency. *Psychological Research*, 1-23.
- Kousta, S. T., Vigliocco, G., Vinson, D. P., Andrews, M., & Del Campo, E. (2011). The representation of abstract words: why emotion matters. *Journal of Experimental Psychology: General*, 140(1), 14.
- Lakhzoum, D., Izaute, M., Ferrand, L. (2020). Intangible feature extraction in the semantic processing of abstract concepts. *61st Annual Meeting of the Psychonomic Society « Virtual Psychonomics »*, 2020, Austin, United States.
- Lupyan, G., & Mirman, D. (2013). Linking language and categorization: Evidence from aphasia. *Cortex*, 49(5), 1187-1194.
- Paivio, A. (1990). *Mental representations: A dual coding approach*. Oxford University Press.
- Schwanenflugel, P. J., Akin, C., & Luh, W. M. (1992). Context availability and the recall of abstract and concrete words. *Memory & Cognition*, 20(1), 96-104.
- Villani, C., Lugli, L., Liuzza, M. T., & Borghì, A. M. (2019). Varieties of abstract concepts and their multiple dimensions. *Language and Cognition*, 11(3), 403-430.
- Wauters, L. N., Tellings, A. E., Van Bon, W. H., & Van Haften, A. W. (2003). Mode of acquisition of word meanings: The viability of a theoretical construct. *Applied Psycholinguistics*, 24(3), 385.

Appendix C

Associate Editor Comments to Author (Dr Rochelle Ackerley):

Comments to the Author:

Further comments have been received from one reviewer and these should be taken into consideration when modifying your paper. The reviewer makes a very good point about the pre-registration of studies and how these rules need to be integrated into your work.

Reviewer comments to Author:

Reviewer: 1

Comments to the Author(s)

This is a revised manuscript, which I have already evaluated previously. I would like to thank the authors for their detailed response to my two concerns raised in the previous review round. In particular, it is commendable that they performed additional control experiments including the semantic relatedness judgment task, which I have previously proposed.

The results from these control studies clearly indicate that for ACs pictures are more difficult to match than for CCs. I agree with the authors' claim that this might be an intrinsic property of ACs. However, in the revised manuscript the new control studies are only described and discussed in the Supplementary Material. It is essential for a proper interpretation of the entire findings of the study that the main text including the abstract reflects the results of the new control studies. For that reason, the discussion of these control studies must be moved from the Supplementary Material to the main text, for instance to the discussion section. The abstract should also reflect this balanced interpretation of the results in terms of conceptual difficulty. The control studies in the Supplementary Material should also be referred to in the Methods section.

We thank the Reviewer for these suggestions. We moved the discussion on the control experiments in the Discussion session, page 21, Lines 652-661, we added some lines into the abstract: Line 30 pag.1, and Lines 42-46 pag.1 and we refer in the Methods section to the control experiments reported in Supplementary Materials, lines: 141-144, pag.5.

Furthermore, the effect of conceptual difficulty of the CCs and ACs on objective and subjective help indices in the guessing task must be assessed by entering RTs to the individual ACs and CCs from the relatedness judgment task as covariate in LMM analyses. It is important to see, whether or not the effect of conceptual category disappears when conceptual difficulty is entered as covariate. If yes, this would slightly alter the interpretation of the results.

We thank the Reviewer for the suggested analysis that might clarify the weight of the conceptual difficulty on the ratings of objective and perceived need of other's help. In the following we try to clarify different analysis options, and their downsides.

First of all, we would like to make it clear that the data of the guessing task and those of the semantic relatedness judgment one come from two different experimental groups and we don't think that using one of the two as a covariate for the other is legitimate.

It looks like the Reviewer is asking us to run a new LMM using the RTs of the semantic relatedness task as a covariate of the subjective and objective helping indices to try and evaluate whether, once the RTs for ACs and CCs are taken into account, the difference in the helping indexes between the two categories would disappear.

Unfortunately, aside from the fact that the data come from two different groups, many methodological problems prevent us from performing the suggested statistical analysis:

- 1) The subjective helping index was not recorded after each trial, but it was recorded after the entire block of abstract or concrete concepts in the guessing concept task; thus, we do not have this index for each word and we cannot associate the RTs of each stimulus in the relatedness task to the average of the subjective index for abstract or concrete concepts.
- 2) For what concerns the objective helping index, instead, the only way to test whether the difficulty of the two categories accounts for the difference in the helping index would be to run an analysis on the stimuli (i.e., evaluate whether the stimuli from the two categories do not differ in terms of their objective helping index once you covariate it with its RTs in the relatedness semantic task) but we think that no LMM could be run anyway since no random factor could be modelled. Indeed, in order to test the hypothesis, we would need to enter as dependent variable the objective helping index calculated from participants' performance at the concept guessing task averaged for each stimulus, and as a predictor, the averaged participants' values of RTs calculated from the semantic relatedness judgment task. However, under these conditions there would be no random factor since the participants are averaged for each stimulus and if we insert the stimulus as the random part, then the number stimuli would be coincident with the number of observations. This would prevent us from running a LMM since the number of levels of each grouping factor (i.e., stimulus) must be smaller than the number of observations, which are again the words.

As a further attempt to address the Reviewer's concern, we performed a correlation between the averaged RTs calculated from the semantic relatedness judgment task and the objective helping index for each stimulus.

The Spearman correlation was significant ($R=.56$ $p=.00032$): the more participants asked for suggestions, the higher were RTs to associate the concept with the images (regardless of the concepts being abstract or concrete). This result confirms that the semantic relatedness judgement task can be considered an additional implicit measure of the conceptual difficulty together with the explicit measure concerning the objective helping index. We thank the Reviewer for the clever inputs that we hope to properly investigate in the future; indeed, the role of the conceptual difficulty is a pivotal issue worth of a better understanding.

We now added this passage in the supplementary materials : Lines 933-939, pag. 35 "Interestingly, a significant Spearman correlation ($R=.56$ $p=.00032$), between the objective helping index from the concept guessing task and the RTs from the semantic relatedness task (only with the stimuli employed in the main experiment), indicates that the more participants asked for suggestions, the higher were RTs to associate the concept with the images. We can conclude that the semantic relatedness judgment task might be considered an additional implicit measure of

the conceptual difficulty together with the explicit measure concerning the objective helping index.”

Finally and importantly: The author’s study is pre-registered on the OSF platform. In order to conform with the Open Science guidelines, it is essential to indicate in the main text, which parts of the methods and analyses deviate from the pre-registration and are therefore exploratory, rather than confirmatory. Therefore, the sentence on p. 5 lines 136-138 must be reworded accordingly because it does not correctly capture all facets of the study: “All the hypotheses, experimental procedures, and data analyses have been specified in a pre-registration <https://osf.io/4tbme> analyses including the covariates can be found in the supplementary materials.”

We thank the Reviewer to ask for this clarification. We now specified in the sentence the confirmatory and the exploratory analyses.

“All the hypotheses, experimental procedures and data analyses have been specified in a pre-registration <https://osf.io/4tbme>. The analyses including the covariates and control experiments to assess the stimuli validity can be found in the supplementary materials. The paragraph named “Stimuli validity check” in Supplementary Materials contains analyses that were not preregistered and are therefore exploratory.”

“

===PREPARING YOUR MANUSCRIPT===

- one version identifying all the changes that have been made (for instance, in coloured highlight, in bold text, or tracked changes);
- a 'clean' version of the new manuscript that incorporates the changes made, but does not highlight them. This version will be used for typesetting if your manuscript is accepted.

===PREPARING YOUR REVISION IN SCHOLARONE===

-- Ensure that your data access statement meets the requirements at <https://royalsociety.org/journals/authors/author-guidelines/#data>. You should ensure that you cite the dataset in your reference list. If you have deposited data etc in the Dryad repository, please include both the 'For publication' link and 'For review' link at this stage.
